# Divergent genomic trajectories predate the origin of animals and fungi

Eduard Ocaña-Pallarès[1,2 ✉], Tom A. Williams[3], David López-Escardó[1,4], Alicia S. Arroyo[1], Jananan S. Pathmanathan[5], Eric Bapteste[6], Denis V. Tikhonenkov[7,8], Patrick J. Keeling[9], Gergely J. Szöllősi[2,10,11] & Iñaki Ruiz-Trillo[1,12,13 ✉]

Animals and fungi have radically distinct morphologies, yet both evolved within the same eukaryotic supergroup: Opisthokonta[1,2]. Here we reconstructed the trajectory of genetic changes that accompanied the origin of Metazoa and Fungi since the divergence of Opisthokonta with a dataset that includes four novel genomes from crucial positions in the Opisthokonta phylogeny. We show that animals arose only after the accumulation of genes functionally important for their multicellularity, a tendency that began in the pre-metazoan ancestors and later accelerated in the metazoan root. By contrast, the pre-fungal ancestors experienced net losses of most functional categories, including those gained in the path to Metazoa. On a broad-scale functional level, fungal genomes contain a higher proportion of metabolic genes and diverged less from the last common ancestor of Opisthokonta than did the gene repertoires of Metazoa. Metazoa and Fungi also show differences regarding gene gain mechanisms. Gene fusions are more prevalent in Metazoa, whereas a larger fraction of gene gains were detected as horizontal gene transfers in Fungi and protists, in agreement with the long-standing idea that transfers would be less relevant in Metazoa due to germline isolation[3–5]. Together, our results indicate that animals and fungi evolved under two contrasting trajectories of genetic change that predated the origin of both groups. The gradual establishment of two clearly differentiated genomic contexts thus set the stage for the emergence of Metazoa and Fungi.

One of the most surprising early insights of molecular phylogenetics was the close evolutionary relationship between animals and fungi[6], which was unexpected because of the enormous differences in their morphology, ecology, life history and behaviour. This relationship has stood the test of time, and now animals and fungi are members of Holozoa and Holomycota, respectively, which are the two major divisions of the eukaryotic supergroup Opisthokonta[1]. Pinpointing how animals and fungi evolved to be so different requires a detailed reconstruction of the evolutionary changes leading up to the two lineages. This demands not only genomic data from diverse animals and fungi but also from the protist opisthokont groups that branch between them (Fig. 1d), which are underrepresented in genomic databases[7].

## Four new genomes of protist opisthokonts

The closest known groups to Metazoa within Holozoa are Choanoflagellatea, Filasterea and Teretosporea (Fig. 1d). Within Holomycota, the closest known groups to Fungi (here defined as the least inclusive clade including Chytridiomycota and Blastocladiomycota based on the absence of phagotrophy in all the members of this clade[8]) are Opisthosporidia (a paraphyletic group[9,10], which in our genomic dataset is represented by *Rozella allomycis* and *Mitosporidium daphniae*−RM clade) and Nucleariidae (Fig. 1d). To improve the limited genome sampling for the protist opisthokont groups[7], we sequenced, assembled and annotated the genomes of three filastereans (*Ministeria vibrans*[11], *Pigoraptor vietnamica*[12] and *Pigoraptor chileana*[12]) and one nucleariid (*Parvularia atlantis*[13]) from metagenomic data produced from cultures of these species (Supplementary Information 1). Given that Filasterea and Nucleariidae were previously represented by only a single whole-genome-sequenced species, the four newly sequenced species represent a substantial increase in the diversity of genomic data available for the protist opisthokont groups (Fig. 1d). This can be expected to minimize the negative impact of poor taxon sampling in ancestral reconstructions (see an example of this issue in Extended Data Fig. 1a).

The four sequenced genomes present high completeness and contiguity metrics, which are in the range of those from the previously sequenced protist opisthokont species (Fig. 23 in Supplementary Information 1). With regard to genome size and gene content metrics, the sequenced

[1]Institut de Biologia Evolutiva (CSIC-Universitat Pompeu Fabra), Barcelona, Spain. [2]Department of Biological Physics, Eötvös Lorand University, Budapest, Hungary. [3]School of Biological Sciences, University of Bristol, Bristol, UK. [4]Ecology of Marine Microbes, Institut de Ciències del Mar (ICM-CSIC), Barcelona, Spain. [5]Equipe AIRE, UMR 7138, Laboratoire Evolution Paris-Seine, Université Pierre et Marie Curie, Paris, France. [6]Institut de Systématique, Evolution, Biodiversité (ISYEB), Sorbonne Université, CNRS, Museum National d'Histoire Naturelle, EPHE, Université des Antilles, Paris, France. [7]Laboratory of Microbiology, Papanin Institute for Biology of Inland Waters, Russian Academy of Sciences, Borok, Russia. [8]AquaBioSafe Laboratory, University of Tyumen, Tyumen, Russia. [9]Department of Botany, University of British Columbia, Vancouver, British Columbia, Canada. [10]MTA-ELTE "Lendület" Evolutionary Genomics Research Group, Budapest, Hungary. [11]Institute of Evolution, Center for Ecological Research, Budapest, Hungary. [12]Departament de Genètica, Microbiologia i Estadística, Facultat de Biologia, Institut de Recerca de la Biodiversitat (IRBio), Universitat de Barcelona (UB), Barcelona, Spain. [13]ICREA, Barcelona, Spain. ✉e-mail: ed3716@gmail.com; inaki.ruiz@ibe.upf-csic.es

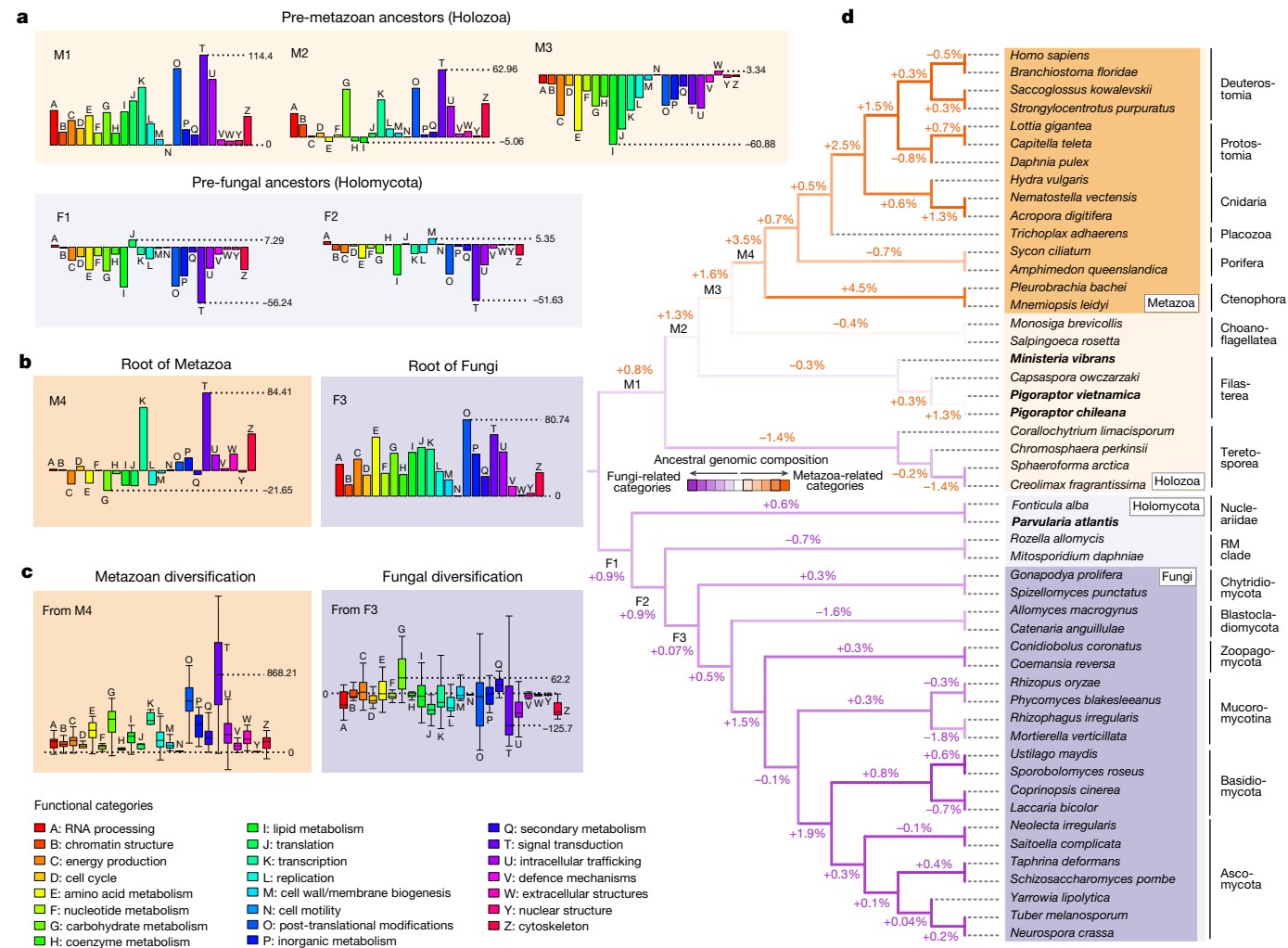

**Fig. 1 | Lineages leading to modern Metazoa and Fungi experienced sharply contrasting trajectories of genetic changes. a,b,** Net gains and losses of 'Cluster of Orthologous Groups' categories with functional information (hereafter referred to as functional categories) since the divergence of Opisthokonta to the emergence of both groups. See Extended Data Fig. 4 for full category names and for information on the other ancestral nodes. **c,** Boxplot distribution of the cumulative net gains and losses of functional categories that occurred in each of the ancestral paths leading to the extant representatives of Metazoa (*n* = 15) and of Fungi (*n* = 21) since the origin of both groups (Supplementary Tables 1 and 2). Outliers are not represented,

but a fully displayed version of **c** is available in Supplementary Fig. 1. Note that, on average, Metazoa tended to accumulate genes for every functional category, whereas only a few categories experienced net gains in the path to modern Fungi. **d,** Changes in functional category composition during the evolution of Opisthokonta, with percentages indicating the magnitude of change in each ancestor (Supplementary Table 3). Metazoa-related and Fungi-related categories are indicated in Fig. 2a. The cladogram shown was reconstructed based on the most supported topologies found for Holozoa and Holomycota in the phylogenetic analyses (Supplementary Information 3). Genomic data were produced for the four species in bold.

## Large differences in gene content

We explored whether the gene contents of Metazoa and Fungi present broad-scale functional differences as this would be indicative that, at some point after the divergence of their last common ancestor, a substantial genetic turnover occurred (that is, the remodelling of the gene content as a result of gene gains and losses, with gains including the origination of novel gene families and the expansion of ancestral

species are not different from most unicellular eukaryotes and fungi (Extended Data Figs. 2 and 3) with the exception of *P. atlantis*. Despite having a compact genome (19.24 Mb), this nucleariid presents 8.58 introns per gene (Extended Data Fig. 3a). This ratio is almost identical to *Homo sapiens*, despite the introns of *P. atlantis* being approximately 86 times shorter (60.67 mean bp size) (Extended Data Fig. 3b), giving it an intron density (approximately four introns per kilobase) more than twice that of any other genome explored (Extended Data Fig. 1b).

families). In a multivariate analysis of the relative genomic representation of each Cluster of Orthologous Groups functional categories[14] (hereafter referred to as functional categories), Metazoa and Fungi cluster separately in the dimension accounting for the largest variance explained (68.1%) (Fig. 2a). Functional categories of signal transduction (T), transcription (K) and extracellular structures (W), which are particularly relevant for animal multicellularity[15,16], are among the most differentially represented in animal genomes (particularly T and W; Extended Data Fig. 5a). Other categories that are more represented in Metazoa include cytoskeleton (Z) and cell motility (N) (Fig. 2a). By contrast, the vast majority of metabolic functional categories (C, E, F, G, H, I and Q; see Fig. 1c) are proportionally more represented in Fungi (Fig. 2a).

## Greater divergence of metazoan gene sets

From an evolutionary perspective, the large genetic differences shown between Metazoa and Fungi might be explained because either both

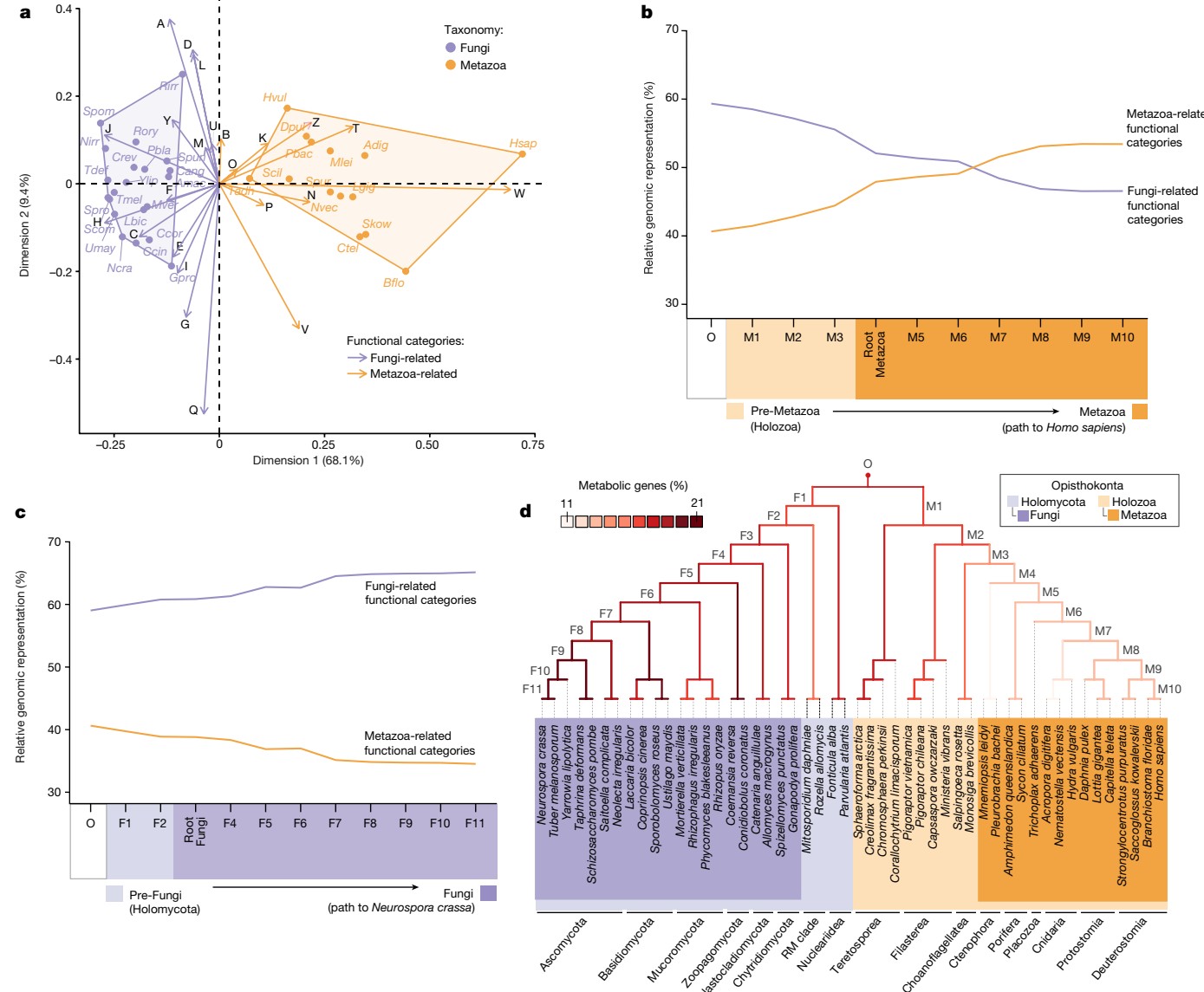

**Fig. 2 | Gradual compositional change at the gene function level predated the origin of Metazoa and Fungi. a**, Correspondence analysis on the functional category compositions of modern metazoan and fungal gene contents (see species names in Supplementary Table 4). *Amphimedon queenslandica* was excluded because its outlier behaviour impairs proper data visualization (Extended Data Fig. 6a). Metazoa and Fungi cluster separately in dimension 1, the axis concentrating the largest fraction of variability (68.1%). Functional categories were grouped as Fungi-related or Metazoa-related from their contribution to dimension 1. **b,c**, Evolution of the functional category compositions in the ancestral paths leading to the species that got the highest scores by the machine learning classifiers that were trained to detect functional category compositions characteristic of Metazoa (**b**) and Fungi (**c**) (Supplementary Table 5). See the functional category composition of each ancestral node in Fig. 1d. **d**, Evolution of metabolic genomic representation in Opisthokonta, measured as the percentage of gene content represented by Kyoto Encyclopedia of Genes and Genomes (KEGG) Orthology Groups related to metabolism (Supplementary Table 3). Fungi have a larger fraction of their gene content involved in metabolism.

or just one of the two groups experienced substantial genetic changes after diverging from their last shared common ancestor. Furthermore, this divergence could either be due to an abrupt genetic turnover in which changes would have occurred specifically in the root of both groups, or by a gradual process in which the preceding ancestors of each group were already accumulating changes in the direction of the differences observed in extant Metazoa and Fungi (Fig. 2a). To distinguish between these alternative scenarios, we took two complementary approaches to reconstruct the tempo and modes of the genetic divergence that occurred. In the first approach, we split the functional categories into two groups based on the results from the multivariate analysis on extant species from Metazoa and from Fungi (Fig. 2a): Metazoa-related or Fungi-related. Then, we computed the

relative representation of each group of functional categories in every ancestral node of Opisthokonta (Fig. 1a) based on the gene contents inferred with our ancestral reconstruction pipeline (see Methods). In the second approach, we trained a series of machine learning classifiers to find their own functional category-based definition based on the gene contents from extant Metazoa and Fungi (see Methods). Then, we scored the ancestral nodes—which were not used to train the classifiers—according to how metazoan-like and fungal-like the relative compositions of functional categories of their inferred gene contents were (Extended Data Fig. 4d).

Not surprisingly, Fungi-related functional categories are more represented in Fungi (particularly in Basidiomycota and Ascomycota groups), but for most of the non-metazoan and non-fungal

opisthokonts, the relative genomic representation of functional categories is more Fungi-like than Metazoa-like (Fig. 1d). As a result, Fungi does not separate from the protist opisthokont groups as distinctly as Metazoa (Extended Data Fig. 6b). These results are consistent with the fact that the machine learning classifiers differentiate the functional category compositions of Metazoa more strongly than those of Fungi (Extended Data Fig. 4d), as shown by the lower probabilities retrieved for the inner nodes of Fungi (43.7% for F3, root of Fungi) than those retrieved for Metazoa (81.7% for M4, root of Metazoa). Together, these results indicate that Metazoa experienced a broader differentiation at the gene function level than Fungi, with fungal gene contents being more similar to those of the protist opisthokonts, including the root of Opisthokonta (Fig. 1d and Extended Data Fig. 6c).

## Gradual process, punctuated acceleration

Our ancestral reconstruction shows the genetic differences between Metazoa and Fungi (Fig. 2a) stemming from a divergence that started early after the split of Opisthokonta and continued up to the origin of the two groups (Fig. 2b,c). In the path to Metazoa, the changes that occurred in the three pre-metazoan ancestors (M1–M3) together account for a contribution of a similar magnitude to shifting the composition of the lineage towards Metazoa-related functional categories than those changes occurred in the metazoan root (3.7% versus 3.5%; Fig. 1d). Among the pre-metazoan ancestors, the changes in M2 and M3 contributed more than the changes in M1 despite both nodes showing fewer net gene gains (Fig. 1a). This is explained because gains in M1 were distributed across a wider set of functional categories, whereas gains in M2 occurred particularly in Metazoa-related functional categories, and the net losses in M3 were more prevalent in Fungi-related functional categories (Fig. 1a). Notwithstanding the contribution of the pre-metazoan ancestors, at the root of Metazoa (M4) there is also evidence for a substantial burst of net gains from a subset of functional categories (Fig. 1b), including transcription (K), signal transduction (T) and extracellular structures (W), which are particularly relevant for the animal multicellular genetic toolkit[15]. Although in the pre-genomic era the animal multicellular genetic toolkit was largely expected to be the outcome of metazoan-specific genetic innovations (that is, gene families that originated at the metazoan root), comparative genomics has revealed orthologues of many toolkit components in the unicellular relatives of animals[15,17–19]. This finding highlighted the importance that the co-option of ancestral gene originations had for multicellularity, although those same studies, as well as more recent studies[19–21], also reported remarkable gene originations at the metazoan root. To quantify what contributed more to the pool of gene families involved in functions that are particularly important for multicellularity (K, T and W), whether pre-metazoan gene originations from Holozoa or those that occurred at the metazoan root, we traced the evolutionary trajectories of these three categories after the divergence of Opisthokonta.

Of gene gains observed at the metazoan root for K, T and W categories, 42.8% correspond to gene families that originated in this same ancestor (M4), whereas 21.2% of gains in M4 correspond to the expansion of gene families that originated in the pre-metazoan holozoan ancestors (Extended Data Fig. 6d). This difference (42.8% to 21.2%) is much greater than the observed for the other functional categories (19.2% to 15.9%), indicating that among the gene gains that occurred at M4, gene originations were particularly relevant for K, T and W at the metazoan root. An inspection of the ancestral contribution to the gene content of *H. sapiens* (Extended Data Fig. 6e) illustrates the same trend: genes from families originated in M4, a single ancestral node, contributed in a similar extent to the ancestral repertoire of the genes involved in K, T and W in *H. sapiens* (mean of 13.9%) than genes from families originated in the three pre-metazoan ancestral nodes (M1–M3) (mean of 12.5%). From this, we conclude that gene originations at

M4 have been quantitatively more important (13.9% versus 12.5%) to functional categories related to animal multicellularity than the gene originations coming from any of the preceding holozoan ancestors. As a result, the metazoan root experienced a substantial increment in the relative genomic representation of K, T and W (+1.35%, +1.16% and +0.35%, respectively, from M3 to M4) (Extended Data Fig. 6f). Notwithstanding this, the tendency towards increasing the relative genomic representation of these functional categories was already ongoing in the pre-metazoan holozoan ancestors (+1.73%, +0.66% and +0.24%, respectively, from O to M3) and hence predated the origin of animals (Extended Data Fig. 6f).

## Main genetic changes in Fungi

Similar to Metazoa, the genetic changes that occurred in the preceding ancestors of Fungi from Holomycota (F1 and F2) contributed more to shifting the gene content (1.8% together)−in this case, towards Fungi-related functional categories−than the root of the group (0.07%) (Figs. 1d and 2c). However, whereas the ancestral path to Metazoa from M1 to M3 accumulated net gains of Metazoa-related functional categories, F1 and F2 did not accumulate gains but rather losses of Metazoa-related functional categories, particularly signal transduction (Fig. 1a).

The two fungal nodes that present the largest compositional shift towards Fungi-related functional categories are, on the one hand, the stem node of Dikarya (Ascomycota + Basidiomycota) (+1.9%; Fig. 1d), which experienced genetic changes that could have predisposed the evolution of complex multicellularity in some members of this group (see Supplementary Information 4), and on the other hand, the last common ancestor of Zoopagomycota, Mucoromycotina and Dikarya (+1.5%), which experienced important morphological adaptations such as the ancestral loss of the flagellum that is characteristic of most fungal groups[22]. On average, and in contrast to animals, Fungi retained gene contents of a similar size to their ancestors and the protist opisthokonts (Extended Data Fig. 7). Still, some fungal nodes showed substantial net gains, particularly the fungal root (F3; Fig. 1b). Similar to the animal root in Holozoa, F3 was the node in Holomycota with the largest fraction of gene gains being explained by gene originations (Extended Data Fig. 8). Nevertheless, the changes seen at the fungal root made a low contribution to the compositional shift of Fungi (0.07%; Fig. 1d) because this node accumulated net gains of both Metazoa and Fungi-related functional categories (Fig. 1b).

The main characteristic of the genetic turnover that occurred in the path to extant Fungi was a specialization towards metabolism (Fig. 2d), whereas animal genomes specialized towards other functional categories (Fig. 2a). In agreement with this, the metazoan root experienced a net loss of metabolic genes (Extended Data Fig. 5d), despite this node presenting an overall net gain of gene content (Fig. 1b), whereas the fungal root experienced net metabolic gene gains (Extended Data Fig. 5c). (Note that an additional supplementary analysis with a dataset that includes transcriptomic data from the aphelid *Paraphelidium tribonemae*[9], which is the closest known group to Fungi, suggests that half of the net gene gains originally detected at the fungal root, including the metabolic ones, could have also predated the origin of Fungi; see Supplementary Fig. 2).

The metabolic changes at the gene content level that we described for the root of Metazoa and Fungi did not become a tendency that continued during the diversification of both groups, as we detected a net accumulation of metabolic genes in Metazoa, but not in Fungi (Extended Data Fig. 5c,d). The larger representation of metabolism in fungal genomes is thus explained because the gene turnover that occurred during the diversification of Fungi benefitted the metabolic over the non-metabolic functions (Fig. 2d). By contrast, Metazoa accumulated more genes of every category, but gains were not particularly biased towards metabolic functions (Fig. 1c).

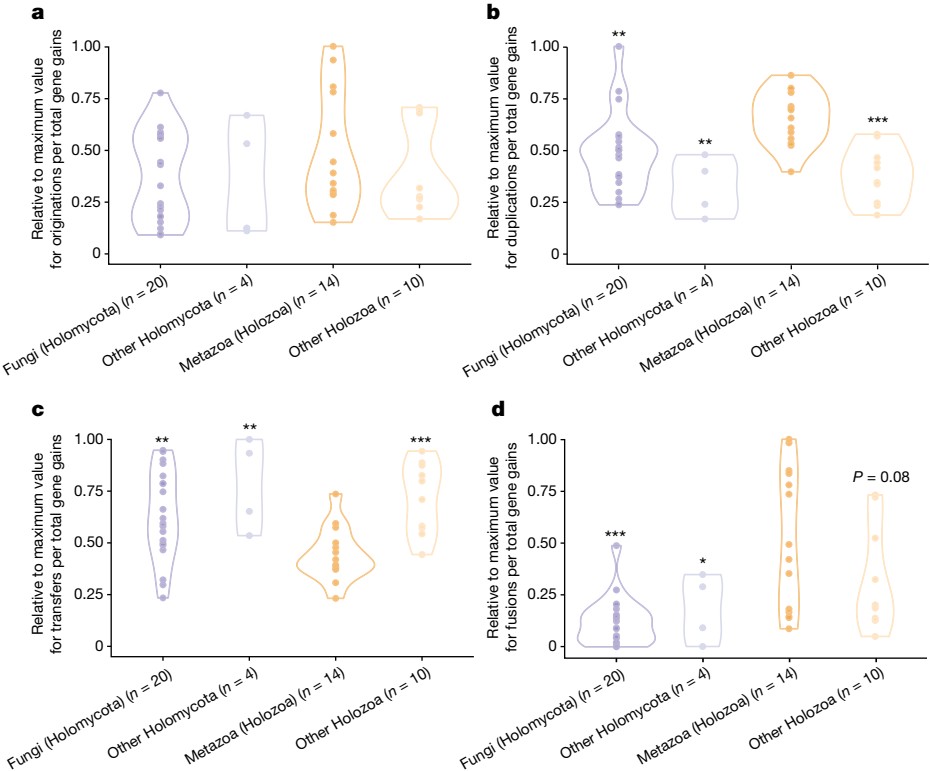

**Fig. 3 | Taxonomic differences in the relative contribution of gene originations, gene duplications, horizontal gene transfers and gene fusions to gene gains. a–d,** Dots correspond to the percentage of gene gains explained by each mechanism in every ancestral lineage of Opisthokonta (Supplementary Table 6; values were normalized to the maximum value found in each plot for a better representation of differences between groups). For every plot, the asterisks indicate the groups that present significantly lower (**b** and **d**) or higher (**c**) distribution of values than Metazoa (Holozoa), according to one-tailed Mann–Whitney $U$-test results. *$P \leq 0.05$, **$P \leq 0.01$ and ***$P \leq 0.001$ (see exact $P$ values in Supplementary Table 6).

## Differences in gene gain mechanisms

Metazoa and Fungi also differ in their preferences among the mechanisms that can be sources of gene gains. Although no significant differences between groups were found in the relative contribution of gene originations to gene gains, gene duplications were found to be significantly more prevalent specifically among metazoan gains (Fig. 3a,b), in accordance with previous studies that highlighted the importance of duplications in the origin and diversification of animals[21]. Besides originations and duplications, the gene tree–species tree reconciliation software[23] used in our ancestral reconstruction framework also estimates putative horizontal gene transfer events as sources of gene gains. Despite being originally described in Bacteria, horizontal gene transfer has been documented across a wide range of eukaryotes and is known to have led to significant functional changes[24–27]. However, the relative contribution of transfers to gene gains in eukaryotes, and whether this contribution is homogeneous across the phylogeny, remain uncertain[28–30]. In this regard, the fact that the reconciliation software recovered a significantly lower fraction of gene gains as being explained by transfers in Metazoa than in Fungi and in the other opisthokonts (Fig. 3c) is compatible with the historical consideration that transfers should contribute less to gene gains in animals due to germline isolation[3–5].

Our ancestral reconstruction pipeline also detects originations that occurred due to gene fusion events. Previous studies[17,18] have described multiple instances of genes in the animal multicellular toolkit that originated through gene fusions (here defined as the merging of partial or complete sequences from older genes). Our results indicate that fusions contributed significantly more to gene gains in Metazoa than in Fungi (Fig. 3d). This is not only explained because Metazoa experienced more gene gains than Fungi (Extended Data Fig. 7), but also because the fraction of originations detected as fusions are also greater in Metazoa (Extended Data Fig. 9). Fusions being less prevalent in Fungi agrees with a previous study that reported a particularly low rate of fusions compared with fissions[31]. Because fusions seem to be particularly relevant sources of transcription and signal transduction genes (Extended Data Fig. 5e,f), this gene gain mechanism could have been more prevalent in Metazoa due to the excess of gains of these two categories (Fig. 1a,b), which are particularly relevant for multicellularity[15].

## Two divergent genomic trajectories

Together, the emerging picture from our ancestral reconstruction indicates that animals and fungi have been evolving under sharply contrasting trajectories of genomic changes that predated the origin of both groups (Fig. 4). Fungal gene contents remained relatively constant in size (Extended Data Fig. 7) and specialized into metabolism (Fig. 2d). By contrast, animals accumulated net gains of most functional categories, although the unequal distribution of gene gains across categories led some categories to increase their relative genomic representation over the others, particularly those that are important for multicellularity (Extended Data Fig. 6f). Although both groups experienced substantial gains and losses during their divergence (Extended Data Fig. 10), the lineage leading to extant Metazoa experienced a larger compositional change in gene function (Fig. 2b,c). As a result, metazoan gene contents are more diverged than the fungal gene contents from those of the other opisthokonts at both the broad-scale functional level and the gene family content level (Extended Data Fig. 6c,g). Given that the latter result is independent of gene function annotation, Metazoa being more differentiated than Fungi from the rest of opisthokonts from a gene content perspective

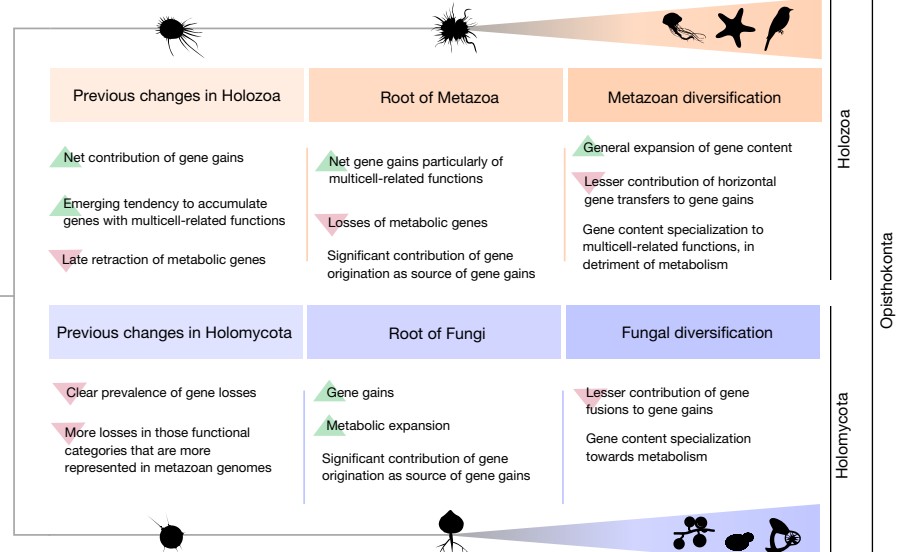

**Fig. 4 | The large genetic differences between modern animals and fungi are the outcome of two contrasting trajectories of genetic changes that preceded the origin of both groups.** These divergent trajectories started immediately after the split of their last common ancestor (Opisthokonta) into Holozoa and Holomycota and continued during the emergence and diversification of Metazoa and Fungi.

is robust to potential inequalities that may exist between groups at the level of biological knowledge or in the availability of functional information. This indeed agrees with the fact that there are more evident morphological discontinuities between protists and animals than between protists and some groups of Fungi[8]. Neither the hypha nor the cell wall characteristic of Fungi, which is also present in some of their protist relatives, are fungal synapomorphies[8]. Only the abandonment of phagotrophy for an osmotrophic lifestyle seems to be a common although not exclusive feature of Fungi[32]. Although animals distinguish from protists from the fact that all of them are multicellular, in Fungi, complex multicellularity is probably the outcome of convergent evolution as it is only found in some particular groups, which present important differences in the genetic contents involved on it[33] (see Supplementary Information 4 for further information on the evolution of multicellularity in Opisthokonta and particularly in Fungi).

From a genomic perspective, the origin of Metazoa and Fungi is better described as a gradual rather than an episodic process given the contribution of their preceding ancestors (M1–M3 and F1–F2) to the cumulative changes at the level of gene function that occurred in the lineages leading to the extant representatives of both groups (Fig. 2b,c). Notwithstanding this, substantial quantitative changes in gene content also occurred concomitantly with the origin of the two groups (Fig. 1b). In particular, the genetic changes at the metazoan root represent an acceleration of a trend that was already ongoing in the pre-metazoan ancestors to accumulate genes of functional categories that are important for animal multicellularity (Extended Data Fig. 6f). These same categories underwent losses in the pre-fungal ancestors (Fig. 1a), situating the immediate ancestors of Fungi and Metazoa in substantially different latent potentials from a genomic perspective. This is especially relevant for the case of animals. Had not animal ancestors experienced a continuous and long-standing evolutionary trajectory that had a compounding effect on the genomic potential for multicellularity, metazoans could not have arisen. The origin of animals may be seen as a drastic evolutionary event, but our taxon-rich analysis shows how the potential for that to happen was generated gradually on a genomic level. Our results illustrate the importance of analysing evolutionary transitions in the light of their evolutionary prehistory.

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

## Methods

### Methodological pipeline for genomic data acquisition

We sequenced a series of culture lines, each including one of the four species of interest (*M. vibrans*, *P. atlantis*, *P. vietnamica* and *P. chileana*). The cultures of *M. vibrans* and *P. atlantis* (formerly *Nuclearia* sp.) were bought in ATCC (*M. vibrans* Tong. ATCC 50519 and *Nuclearia* sp. ATCC 50694, respectively). The cultures of *P. vietnamica* (formerly Opistho-1) and *P. chileana* (formerly Opistho-2) descend from the environmental isolates (*P. vietnamica* from a Freshwater Lake, Vietnam; and *P. chileana* from freshwater temporary water body, Chile) used in ref. [12]. As expected, the starting cultures included an uncertain fraction of contaminant species. In particular, the cultures of *M. vibrans* and *P. atlantis* included an uncertain diversity of bacterial contamination, whereas the cultures of each *Pigoraptor* species also included contamination from the eukaryote *Parabodo caudatus*. The sequenced metagenomic data were submitted to a bioinformatic decontamination pipeline that consisted of two to three rounds of detection and removal of contaminant fragments based on taxonomic and tetranucleotide composition information. All steps were thoroughly supervised to maximize the retention of bona fide genomic fragments from our species of interest and the removal of contaminant sequences. Decontaminated genomes were annotated combining both RNA sequencing-based BRAKER1 v1.9 (ref. [34]) and PASA v2.0.2 (ref. [35]) automatic annotation pipelines, the results of which were processed to correct erroneous gene predictions that might lead to the inference of false gene fusions. See Supplementary Information 1 for a detailed explanation about the nature of the sequenced data and the decontamination and genome annotation processes (see Fig. 1 in Supplementary Information 1 for a summary illustration).

### Clustering sequences into orthogroups

A dataset of 1,463,920 protein sequences from 83 eukaryotic species, 59 from Opisthokonta (including the four genomes produced) and 24 from other eukaryotic groups, was constructed (draft_euk_db; see Supplementary Table 4). Protein sequences were aligned all-against-all using BLASTp[36] v2.5 [-seg yes, -soft_masking true, -evalue 1e-3]. On the basis of the alignments, proteins were clustered into orthogroups (OGs) with OrthoFinder[37] v2.7 [-I2]. We treat OGs as proxies of gene families. The OGs produced by OrthoFinder were processed with the MAPBOS pipeline to fix protein domain heterogeneity problems that would compromise downstream analyses (see Supplementary Information 2 for a discussion of this issue, and for an explanation of the algorithm that we developed to correct it).

### Species tree reconstruction

Ancestral gene contents were inferred by means of a gene tree–species tree reconciliation software. We thus needed to reconstruct a phylogenetic tree for every gene family and a species tree of the whole eukaryotic supergroup Opisthokonta. The results from the species tree reconstruction analyses are available in Supplementary Information 3. We first selected 342 OGs present in >77% of draft_euk_db taxa and with no more than an average of 1.16 copies per taxa. We measured alignment instability of the 342 OGs using COS.pl and msa_set_score v2.02, which are based on the Heads-or-Tails approach[38,39], keeping only those OGs with >0.70 mean column score (MCs). We manually curated the 69 OGs that survived to this filter by performing individual phylogenies for each one, using MAFFT[40] v7.123b [-einsi] for sequence alignment, trimAl[41] v1.4.rev15 [-gappyout] for alignment trimming and IQ-TREE[42] v1.6.7 for maximum-likelihood (ML) phylogenetic inference, using ModelFinder[43] for model selection. Three of these 69 OGs were discarded because the topology was strongly in disagreement with the expected species topology. For the remaining 66 OGs (hereafter referred to as the MCs70 dataset), we removed sequences whose branching pattern suggested that they were most likely misclassified

as OG members. In addition, to keep only one sequence per taxon in every OG, for inparalogue cases, we kept the least divergent sequence according to branch length. We removed a total of 630 sequences from the MCs70 dataset, including likely misclassified OG members but also contaminant sequences. Most contamination cases found correspond to contamination from Stramenopiles in the proteome of *Syssomonas multiformis*, probably from *Spumella* sp.[12]. However, we also detected *Pirum gemmata* contamination in the proteome of *Abeoforma whisleri*, and few from *Ichthyophonus hoferi* in *Sphaerothecum destruens*, indicating cross-contamination problems between these ichthyosporeans datasets. Still, these cases of contamination neither affected the phylogenetic inference, as they were removed during the screening, nor the downstream analyses, as these species were only used for species tree reconstruction purposes.

We created two distinct versions of the MCs70 dataset: the first dataset including all sequences from Holozoa (ingroup) and from three Holomycota taxa (outgroup) (Holozoa MCs70), and the second dataset including all sequences from Holomyoca (ingroup) and from three Holozoa taxa (outgroup) (Holomycota MCs70). An alignment supermatrix was created for each dataset, first aligning and trimming each OG per separate [MAFFT -einsi, trimAl -gappyout], and later concatenating the alignments into a supermatrix (Holozoa MCs70: 37 taxa, 17,475 sites and 9.27% of missing data; Holomycota MCs70: 28 taxa, 17,409 sites and 7.81% of missing data). We constructed a phylogenetic tree for both MCs70 datasets using ML and Bayesian inference. ML inferences were done with IQ-TREE, and the models chosen for Holozoa and Holomycota MCs70 datasets were LG+C50+F+R7 and LG+C30+F+R6, respectively. Despite ModelFinder suggesting the usage of C60 (ref. [44]) for Holomycota MCs70, we used mixture models with fewer profiles to avoid potential model overfitting, as some optimized mixture weights were estimated close to zero. Nodal supports for the ML trees consisted of 1,000 IQ-TREE ultrafast bootstrap replicates (UFBoot) and 100 standard non-parametric bootstrap replicates. Non-parametric bootstraps were computed under the PMSF model[45]. We used the previously inferred ML trees as guide trees to infer mixture model parameters and site-specific frequency profiles, as implemented in IQ-TREE v1.6.7. Bayesian phylogenies were done under the CAT+GTR+Gamma(4) model in PhyloBayes-MPI[46] v1.8. Two chains were run for Holozoa MCs70 and for Holomycota MCs70 supermatrices, and convergence was assessed using the bpcomp and tracecomp programs in the PhyloBayes-MPI package. Consensus trees were built when the maximum between chain discrepancy in bipartition frequencies fell below 0.1 (burn-in 33%). We also performed three additional analyses (increasing number of positions in the supermatrix, compositional recoding and fastest-evolving sites removal) to test the robustness of the topological relationships found (see Supplementary Information 3).

### Incorporation of prokaryotic homologues into the OGs

We incorporated prokaryotic homologues into the clusters before the MAPBOS processing step. For the incorporation of prokaryotic (and viral) homologues into the clusters, we first used DIAMOND[47] v0.8.22.84 [--more-sensitive, -e 1e-05] to align all eukaryotic sequences from euk_db (a subset of draft_euk_db, which includes the species labelled in bold in Supplementary Table 4) to a database including 8,231,104 bacterial, 331,476 archaeal and 20,955 viral from Uniprot reference proteomes (release 2016_02; prok_db) (forward alignment approach). The aligned sequences from prok_db were aligned back against euk_db sequences (reverse alignment approach). Hits with a query and target alignment coverages lower than 75% were discarded, as well as hits in which the best-scoring euk_db target of a given prok_db query was a member of a distinct cluster than the best-scoring euk_db query for that prok_db sequence in the forward alignment. After discarding the hits not satisfying these conditions, we incorporated into the clusters only the best-scoring prok_db query of each euk_db target sequence (that is, if a cluster has 300 sequences and the best-scoring query of all

them was the same prok_db sequence, only that sequence will be incorporated into the cluster, which will then have 300 euk_db sequences and 1 prok_db sequence). Prok_db sequences were incorporated into OrthoFinder -I 2 clusters before these were processed by the MAPBOS pipeline (Supplementary Information 3). After MAPBOS, clusters included 1,117,614 eukaryotic sequences and 58,017 non-eukaryotic sequences (53,168, 4,301 and 548 from Bacteria, Archaea and viruses, respectively). All these 1,175,631 sequences were distributed among 413,445 clusters, 370,686 of which are singletons. Among eukaryotic sequences, on a taxonomic level, clusters included sequences mostly from Opisthokonta (50 species), but also from 18 representatives of other major eukaryotic groups (euk_db dataset).

### Gene tree inference and gene tree–species tree reconciliation analyses

We submitted every post-MAPBOS OGs (or clusters) to a gene tree inference pipeline, consisting of using MAFFT-linsi for the alignment step, trimAl [−gappyout] for alignment trimming and IQ-TREE for the phylogenetic inference. In particular, IQ-TREE was run using the LG+G4 model and sampling 1,000 optimized [-bnni] UFBoot replicates for every gene tree.

For the gene tree–species tree reconciliation analyses, we used ALEml_undated from ALE v0.4 (https://github.com/ssolo/ALE). ALEml_undated requires a distribution of phylogenetic trees for every gene family (the UFBoot replicates in our case) and a species tree. The Opisthokonta fraction of the species tree consisted of the most favoured topology according to our analyses, which only included Opisthokonta taxa (Fig. 1 in Supplementary Information 3). The phylogenetic relationships between the non-Opisthokonta taxa were directly determined from a consensus of currently available bibliographical references[48–56] (all euk_db species were included in the reconciliation analyses). Reconciliation analyses also incorporated non-eukaryotic sequences (see above), which, for practical reasons, were assigned to the same terminal node in the species tree (named 'Prokaryotes' in Fig. 7 in Supplementary Information 3). Eukaryotes with only transcriptomic or poor-quality genomic data were excluded from the reconciliation analyses (those labelled in grey in Fig. 1 in Supplementary Information 3). Note that the inclusion of transcriptomic data would have been particularly problematic to our study for the following reasons: (1) gene content predictions from transcriptomic tend to present inflated gene counts. For example, the proteomes that were previously produced based solely on transcriptomic data for *P. atlantis*[2] and for *P. vietnamica* and *P. chileana*[12] include much more sequences (29,620, 46,018 and 37,783) than the proteomes that we predicted from the genome sequences of these species (9,028, 14,822 and 14,510), with the genome-based proteomes showing even better completeness metrics (Fig. 23 in Supplementary Information 1). Inflated gene counts are expected to produce an excess of duplication inferences in the reconciliations, whereas (2) unexpressed genes may be confused by gene losses. (3) Transcriptomes are harder to decontaminate due to the lack of genomic context information regarding neighbouring genes, intron sequences or compositional features of the coding sequence, whereas (4) those sequences predicted from partial isoforms are expected to lead to inaccuracies to the software used to detect gene fusions (see below). (5) Accurate gene contents were also important given that the reconciliation software used (see above) infers the values for parameters such as gene duplication and loss rates from the data.

### Inference of gene fusion events

We used CompositeSearch[57] to identify composite gene families, that is, families of genes whose protein sequence is composed by fractions—for example, protein domains—that are separately found in other, component, gene families. CompositeSearch requires as input all-against-all sequence alignments, for which we used the same BLASTp results used for OrthoFinder (see above), although alignment hits corresponding to draft_euk_db species not represented in euk_db were removed. Before being used as input for CompositeSearch, BLASTp results were preprocessed with cleanBlastp (included in CompositeSearch) to retain only the hit with the highest score among all hits involving the same query–target pair. CompositeSearch was run with the default parameters and forcing the software [-f] to work on the clusters resulting from the processing of the OG from OrthoFinder by the MAPBOS pipeline. Families with only one sequence were discarded as potential components [-y]. Prok_db sequences were not included in composite inferences as alignments between prok_db and euk_db sequences were done with DIAMOND instead of BLASTp due to computational time limitations. Because we work at the gene family level (clusters), we only considered as composites those clusters in which >50% of members were detected as composite sequences. This includes 48,066 clusters, 3,229 of which are not singletons.

CompositeSearch detects as a composite any sequence that matches with distinct subsets of sequences (components, from other OGs) in different regions of its sequence. Whereas fusion events may lead to composite sequences, not all sequences detected as composites necessarily originated from a gene fusion process. For example, a sequence found to be composite by the software could have originated de novo in a given ancestral lineage (gene X–domains A and B), and then, in a descendant lineage, that gene could have been split into two separate genes (gene Y–domain A and gene Z–domain B). In such a case of gene fission, the software would detect the gene X as a composite because some part of the sequence would be aligned by the gene Y (first component) and the other by the gene Z (second component). To retain only bona fide fusion composite sequences, we only considered those composite sequences in which all their components were inferred to have a more ancestral origin than the composite. This was done to minimize the false-positive inferences of fusions, at the expense of losing potential fusion events in which, for example, both the composite and the components may have originated in the same node of the phylogeny.

### Functional annotation of sequences and OGs

Protein domain architectures of euk_db sequences and of prok_db captured sequences (see above) were determined with PfamScan[58] using Pfam A v29. Cluster of Orthologous Groups functional categories (functional categories) and KEGG Orthology Groups (KOs)[59] were annotated to euk_db sequences with eggNOG-mapper[60] v1.0.3-3-g3e22728, using DIAMOND for the alignments of euk_db sequences against the eggNOG database (the functional category 'S: unknown function' was ignored as it does not include functional information). Once sequences were annotated, the functional categories and KO annotations of every cluster were determined by averaging the annotations of the corresponding cluster members. For example, if a cluster includes two sequences (SeqA and SeqB), and SeqA was annotated with the functional category K and SeqB with the functional categories B and K, that cluster would be annotated as 0.75K and 0.25B (0.5K from SeqA + 0.25K from SeqB, and 0.25B from SeqB).

### Inference of gains, losses and counts of functional categories and metabolic gene contents

From the reconciliation analyses (see 'Gene tree inference and reconciliation analyses'), we retrieved the number of gains, losses and gene contents of every OG in every node in the phylogeny. For every given node, we determined the absolute representation of all functional categories by crossing the information between the number of copies of every OG in the node and the relative representation of every functional category among the functional information of the OGs. The same was done to determine the KO contents of every node. The percentage of metabolic genes of every node was determined by dividing the number of KOs with metabolic annotations by the total number of genes in the node (besides KOs belonging to the 'metabolic category', those

belonging to the category 'membrane transport' were also considered as metabolic genes). The relative representation of every functional category in every node was determined by dividing the absolute value of every category in the node by the sum of the absolute values of all functional categories in the node. Gains and losses of functional categories and KOs were determined by comparing the contents of every node with those of its immediately preceding node.

## Statistical analyses

Statistical analyses were carried out either in Python, mainly with the libraries Pandas[61] and NumPy[62], or in R. All descriptive statistics plots (with the exception of those including phylogenetic trees, which were constructed with ITOL[63]) were done in R, particularly with the ggplot2 package[64]. Mann–Whitney $U$-tests (one-tailed) were done in Python with SciPy[65] (scipy.stats.mannwhitneyu). More specific statistical analyses are detailed below.

## Correspondence analyses of relative functional category compositions

The relative genomic representation of functional categories are examples of compositional data (CoDa)[66], in which every column (a functional category) is represented by a relative fraction and the sum of all values is the same for every row (genome). Owing to the fact that no orthogonality and collinearity are properties of CoDa, most commonly used multivariate analyses techniques such as principal component analyses are unappropriated for CoDa analyses and alternatives such as correspondence analyses are recommended instead[66]. Correspondence analyses were done in R[67] with FactoMiner package[68] and the plots were constructed with the factoextra package[69].

## Machine learning classifiers

For the classifiers of metazoan and fungal functional category compositions, we benchmarked five widely used learning models: logistic regression, $k$-nearest neighbours classifier, support vector classifier, Random Forest and artificial neural network, fine-tuning in every case the model hyperparameters using fivefold cross-validation. In total, we generated two classifiers for every learning model: one trained to distinguish between the functional category compositions of metazoan versus the other terminal nodes in Opisthokonta; and another doing the same but for Fungi instead of Metazoa. Relative functional category compositions were not used as features to train the model by the fact that they are correlated between them. Instead, the models were trained with the components retrieved from the correspondence analyses on the relative functional category compositions of opisthokont terminal nodes (relative compositions were computed excluding the S 'unknown function' category and doing first a column-wise and then a row-wise normalization before correspondence analyses was performed). Once models were trained, we computed the probability of belonging to the given class (Metazoa or Fungi, depending on the model) for every opisthokont node, including both terminal (used for model training) and internal (not used for model training) (see values in Supplementary Table 5). The probabilities represented in Extended Data Fig. 4d correspond to a weighted average over the probabilities retrieved from every classifier (excluding logistic regression for being in disagreement and showing worse predictions than the other classifiers). The weights were determined in the following manner: for every node, the average probability was computed, and then we computed the variance of the four models with respect to that averages. The weight of every model corresponds to the inverse of the relative variance of that model divided by the sum of the variances of the four models. The code is available at https://doi.org/10.6084/m9.figshare.13140191. v1 ('fungiMetazoa_predModels' in Code.300322.zip). We expect the predictors to capture the genomic compositional features well, as, for example, in the case of Metazoa, *Trichoplax adherens*, the animal with the lowest degree of phenotypic complexity among the sampled species, is the node with lowest probability (Extended Data Fig. 4d). All of these analyses were carried out in Python using packages from Sci-kit learn[70], TensorFlow[71] and Keras[72] libraries.

## Reporting summary

Further information on research design is available in the Nature Research Reporting Summary linked to this article.

## Data availability

The raw sequence data and assembled genomes generated in this study have been deposited in the European Nucleotide Archive (ENA) at EMBL-EBI under accession number PRJEB52884 (https://www.ebi. ac.uk/ena/browser/view/PRJEB52884). The genome assemblies are also available in figshare (https://doi.org/10.6084/m9.figshare.19895962. v1). Protein sequences of the species used in this study were downloaded from the GenBank public databases (https://www.ncbi.nlm. nih.gov/protein/), Uniprot (https://www.uniprot.org/), JGI genome database (https://genome.jgi.doe.gov/portal/) and Ensembl genomes (https://www.ensembl.org). The following specific databases were also used in this study: Pfam A v29 (https://pfam.xfam.org/), EggNOG emapperdb-4.5.1 (http://eggnog5.embl.de) and UniProt reference proteomes release 2016_02 (https://www.uniprot.org/). The supporting data files of this study are available in the following repository: https:// doi.org/10.6084/m9.figshare.13140191.v1.

## Code availability

The most relevant custom code developed for this study (the MAPBOS pipeline and the machine learning classifiers) is available at https://doi. org/10.5281/zenodo.6586559.

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

**Acknowledgements** E.O.-P. was supported by a predoctoral FPI grant from MINECO (BES-2015-072241) and by ESF Investing in your future. E.O.-P., D.L-E., A.S.A. and I.R.-T. received funding from the European Research Council under the European Union's Seventh Framework Programme (FP7-2007-2013) (Grant agreement No. 616960) and also from grants (BFU2014-57779-P and PID2020-120609GB-I00) by MCIN/AEI/10.13039/501100011033 and 'ERDF A way of making Europe'. E.O.-P. and G.J.Sz. received funding from the European Research Council under the European Union's Horizon 2020 research and innovation programme (Grant agreement No. 714774). T.A.W. was supported by a Royal Society University Research Fellowship (URF\R\201024) and NERC standard grant NE/P00251X/1. This work was supported by the Gordon and Betty Moore Foundation through grant GBMF9741 to T.A.W. and G.J.Sz. J.S.P. and E.B. received funding from the European Research Council under the European Union's Seventh Framework Programme (FP7-2007-2013) (Grant agreement No. 615274). D.V.T. and cell culturing were supported by the Russian Science Foundation grant no. 18-14-00239, https://rscf.ru/project/18-14-00239/. Culture of *P. vietnamica* was obtained as the result of field work in Vietnam as part of the project 'Ecolan 3.2' of the Russian–Vietnam Tropical Center. P.J.K. is supported by an Investigator Award from the Gordon and Betty Moore Foundation (https://doi.org/10.37807/GBMF9201). We thank the CRG/UPF FACS Unit, the CRG Genomics Unit and M. Antó-Subirats for technical assistance; D. J. Richter, M. M. Leger and I. Patten for the feedback provided on the manuscript; and M. J. Greenacre for the feedback provided on multivariate statistics.

**Author contributions** E.O.-P. conceptualized the study and wrote the draft of the manuscript under the supervision of I.R.-T. E.O.-P., D.L.-E. and A.S.A. generated the material for sequencing. E.O.-P. and A.S.A. made the figures for the manuscript. E.O.-P. performed all bioinformatic analyses (unless those specified below). T.A.W. performed the gene tree–species tree reconciliation analyses and the Bayesian species tree reconstruction, and provided feedback about the project. G.J.Sz. contributed substantially to reviewing the manuscript and providing feedback about the project. J.S.P. and E.B. adapted the software of CompositeSearch and provided feedback about the project. D.V.T. and P.J.K. provided polyxenic cultures from *P. vietnamica*, *P. chileana* and *P. caudatus*. All authors contributed to the review of the manuscript before submission for publication and approved the final version.

**Competing interests** The authors declare no competing interests.

**Additional information**
**Correspondence and requests for materials** should be addressed to Eduard Ocaña-Pallarès or Iñaki Ruiz-Trillo.

A

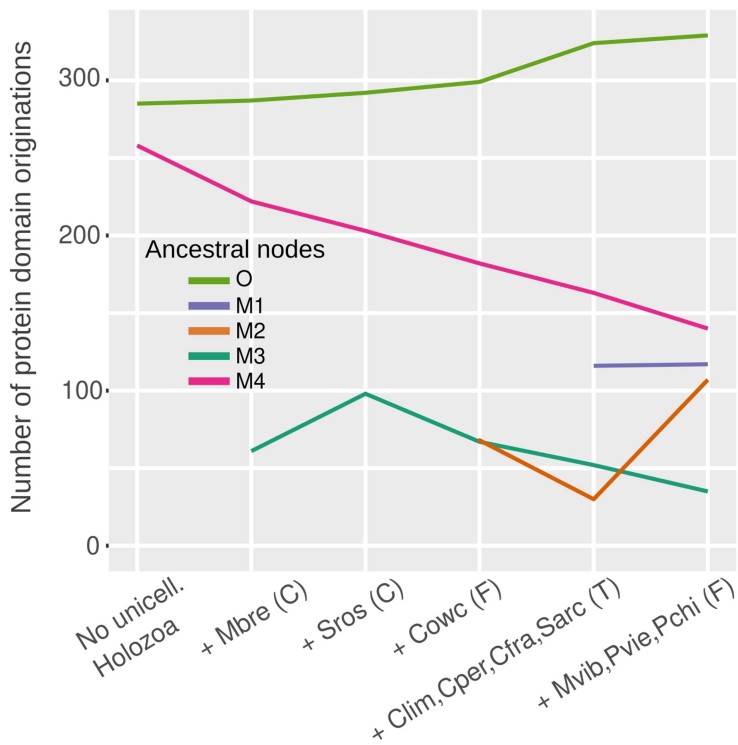

**Taxon sampling and ancestral protein domain originations**

*Y-axis:* Number of protein domain originations

*Legend — Ancestral nodes:*
- O (green)
- M1 (purple)
- M2 (orange)
- M3 (teal)
- M4 (pink)

*X-axis categories:* No unicell. Holozoa; + Mbre (C); + Sros (C); + Cowc (F); + Clim,Cper,Cfra,Sarc (T); + Mvib,Pvie,Pchi (F)

B

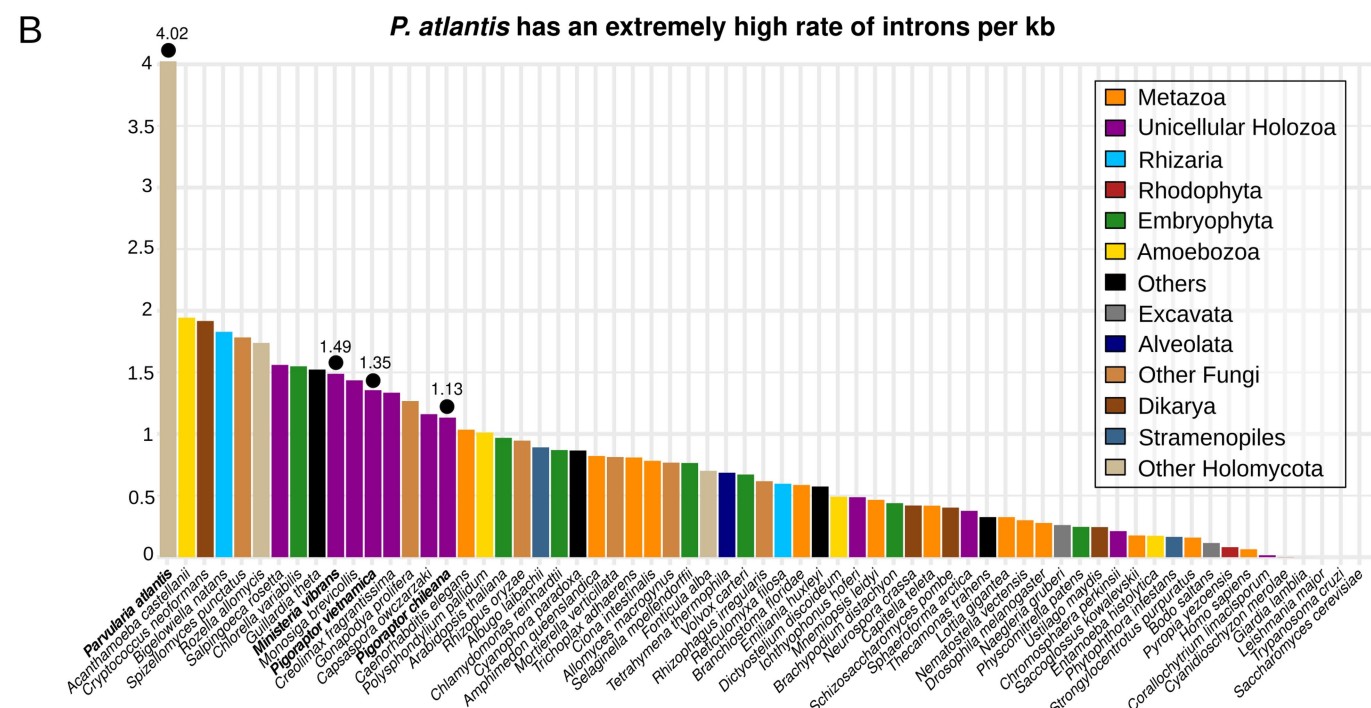

***P. atlantis* has an extremely high rate of introns per kb**

*Legend:*
- Metazoa
- Unicellular Holozoa
- Rhizaria
- Rhodophyta
- Embryophyta
- Amoebozoa
- Others
- Excavata
- Alveolata
- Other Fungi
- Dikarya
- Stramenopiles
- Other Holomycota

**Extended Data Fig. 1 | The importance of taxon sampling in ancestral gene content reconstructions and intron density across eukaryotes. (A)** Influence of taxon sampling in the ancestral reconstruction of protein domains innovations (Pfam domains). Note that with the addition of taxon sampling from unicellular relatives of animals (Choanoflagellatea -C-, Filasterea -F-, Teretosporea -T-), the number of pre-metazoan protein domain originations increase at the expense of originations that were originally detected at M4 in the 'No unicell. Holozoa' condition. The origin of every protein domain was inferred at the last common ancestor of all the species in which the domain is represented. This analysis was carried out with the taxon sampling euk_db,

first excluding all representatives from C, F and T groups ('No unicell. Holozoa'), and then progressively adding data from these groups in a chronological order corresponding to when the genomic data from the representatives of these groups became publicly available. Ancestral node abbreviations: M4 = last common ancestor (LCA) of Metazoa. M3 = LCA of Choanoflagellatea and M4. M2 = LCA of Filasterea and M3. M1 = LCA of Teretosporea and M2. O = LCA of Opisthokonta. (See Fig. 1d for an illustration of the phylogenetic context of these ancestral nodes). (**B**) Distribution of introns per kb in an eukaryotic dataset including the four genomes sequenced for this manuscript as well as the metrics included in the Fig. 1–source data 1 of ref. [18].

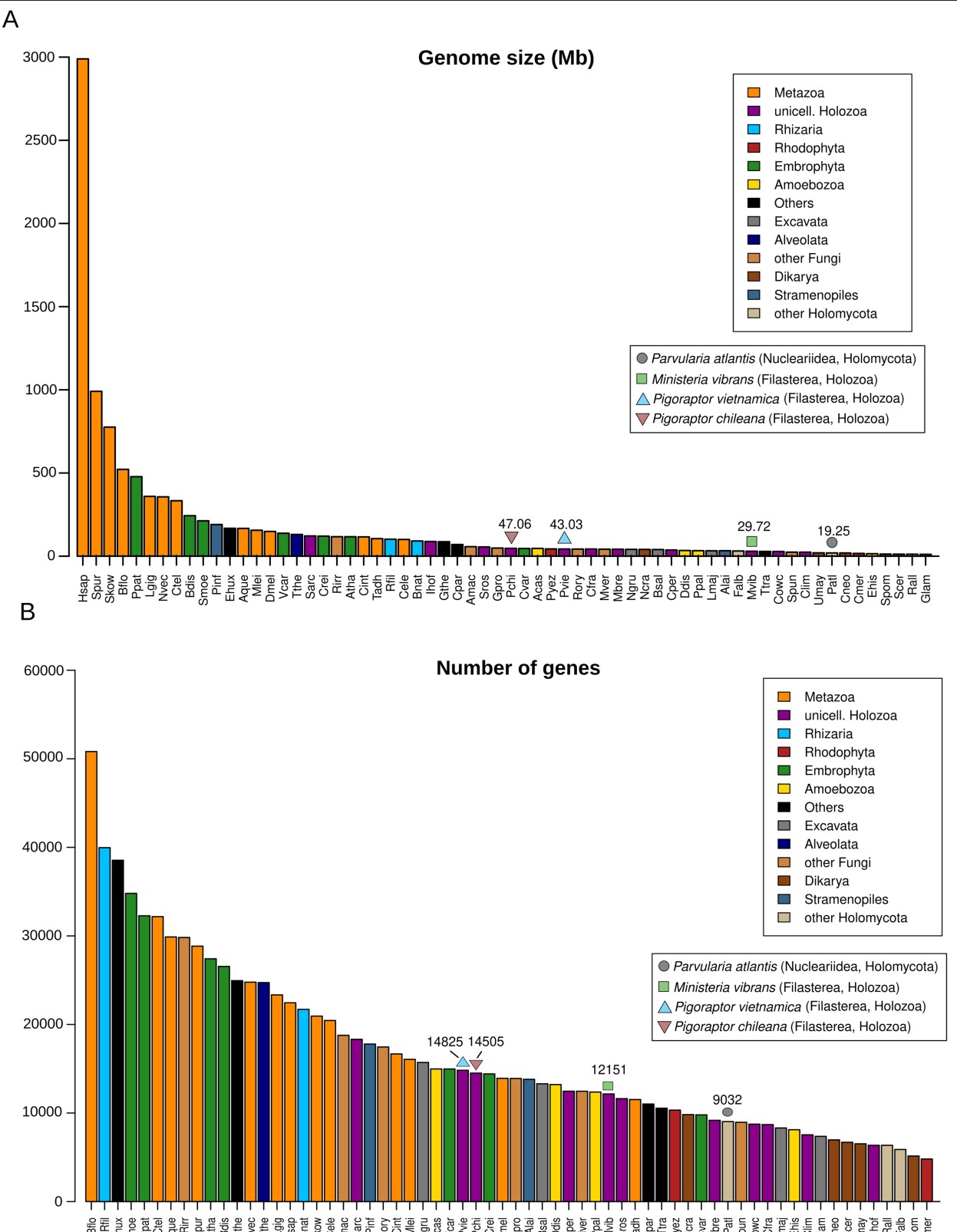

**Extended Data Fig. 2 | Genome size and gene count metrics in eukaryotes.** Distrubtion of (**A**) 'Genome size (Mb)' and (**B**) 'Number of genes' in an eukaryotic dataset including the four genomes produced as well as the metrics included in the Fig. 1—source data 1 of ref. [18].

## A

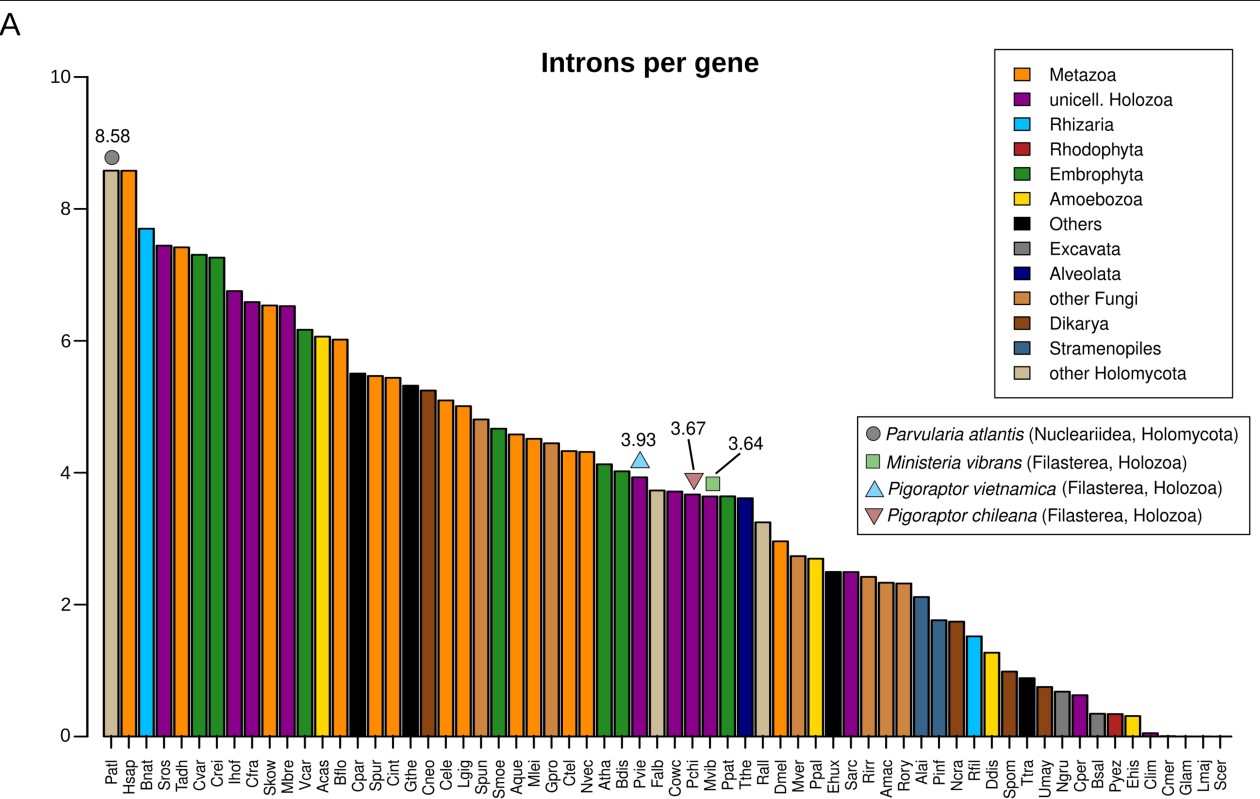

## B

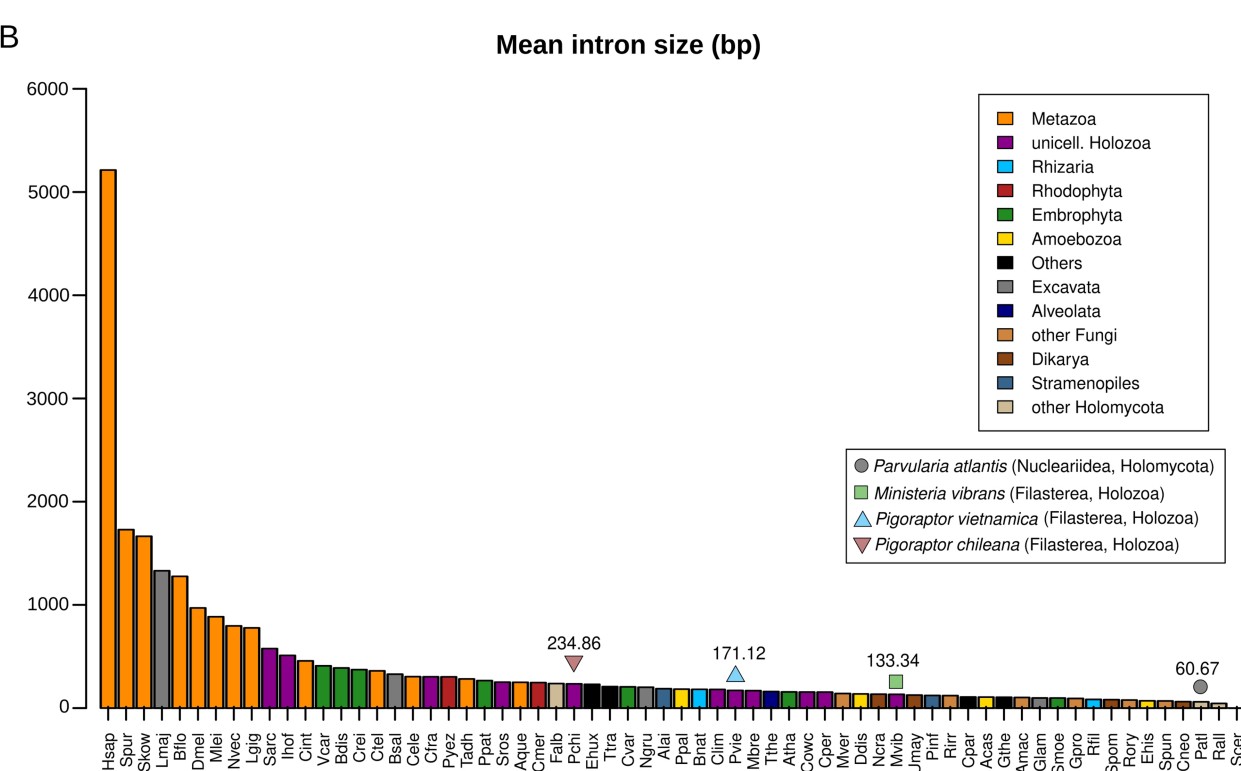

**Extended Data Fig. 3 | Intron per gene and mean intro size metrics in eukaryotes.** Distrubtion of (**A**) 'Introns per gene' and (**B**) 'Mean intron size (bp)' in an eukaryotic dataset including the four genomes produced as well as the metrics included in the Fig. 1–source data 1 of ref. [18]. Whereas a potential loss of non-coding regions in the *P. atlantis* genome during the metagenome decontamination could have led to an underestimation of the genome size metric, the high ratio of introns per gene and the small size of introns found strongly suggests that the intron-richness of this nucleariid is not an artefactual result.

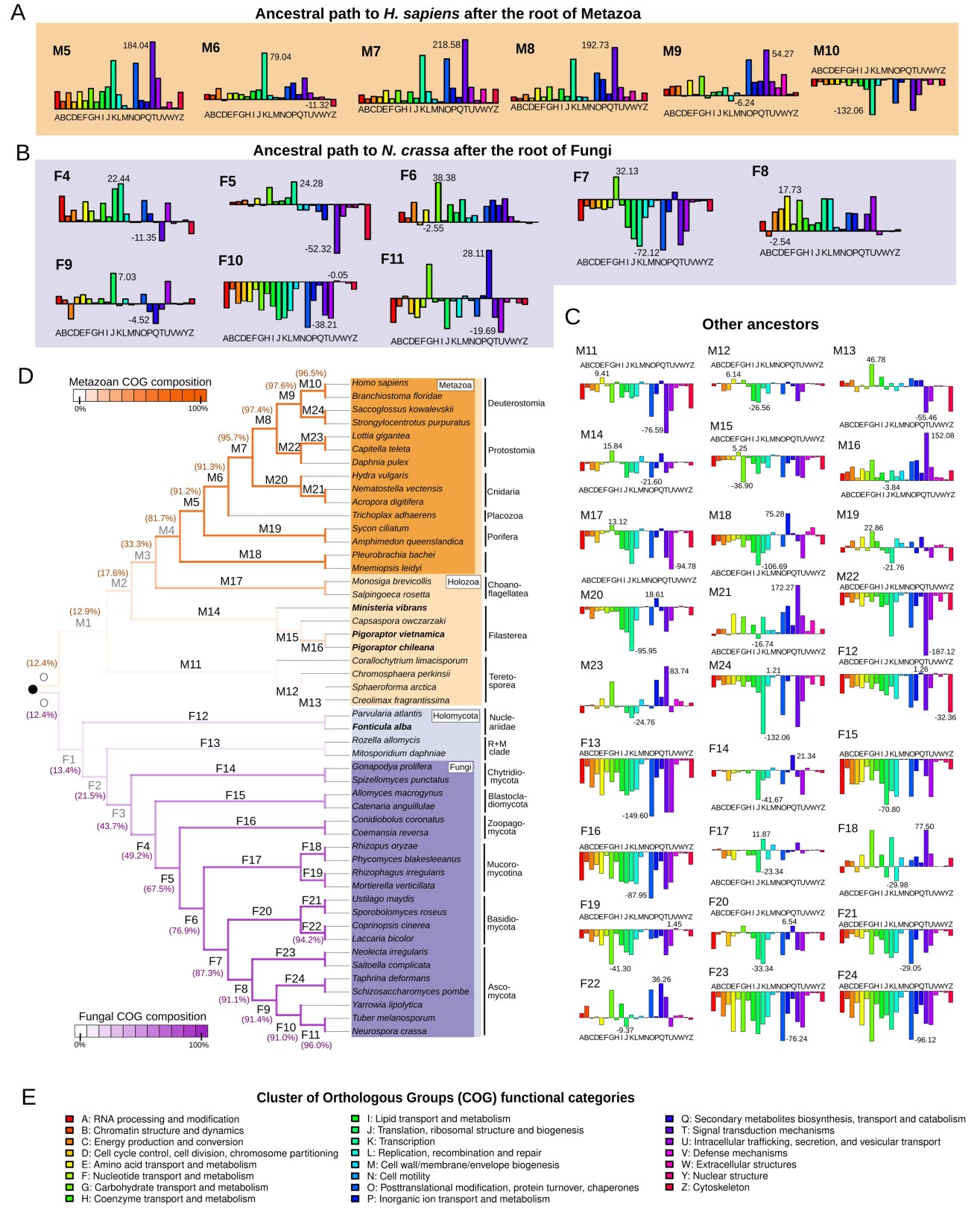

**Extended Data Fig. 4 | Evolution of functional category composition in Opisthokonta.** (**A**–**C**) Net gains and losses of functional categories in those ancestral nodes that are not represented in Fig. 1. (**D**) Consensus phylogeny of Opisthokonta as reconstructed from the phylogenetic analyses (Supplementary Information 3). Genomic data was produced for the four species in bold. Branch colors correspond to the weighted average probability retrieved for every ancestor (internal branches) by the machine-learning

classifiers that were trained to detect differential COG-compositional features of extant Metazoa and of Fungi (see Methods). Branch colors in the Holozoa clade represent the weighted averages from the metazoan predictors, and in the Holomycota clade the weighted average from the fungal predictors (Supplementary Table 5). (**E**) Cluster of Orthologous Groups (COG) categories with functional information (referred to as functional categories along the manuscript).

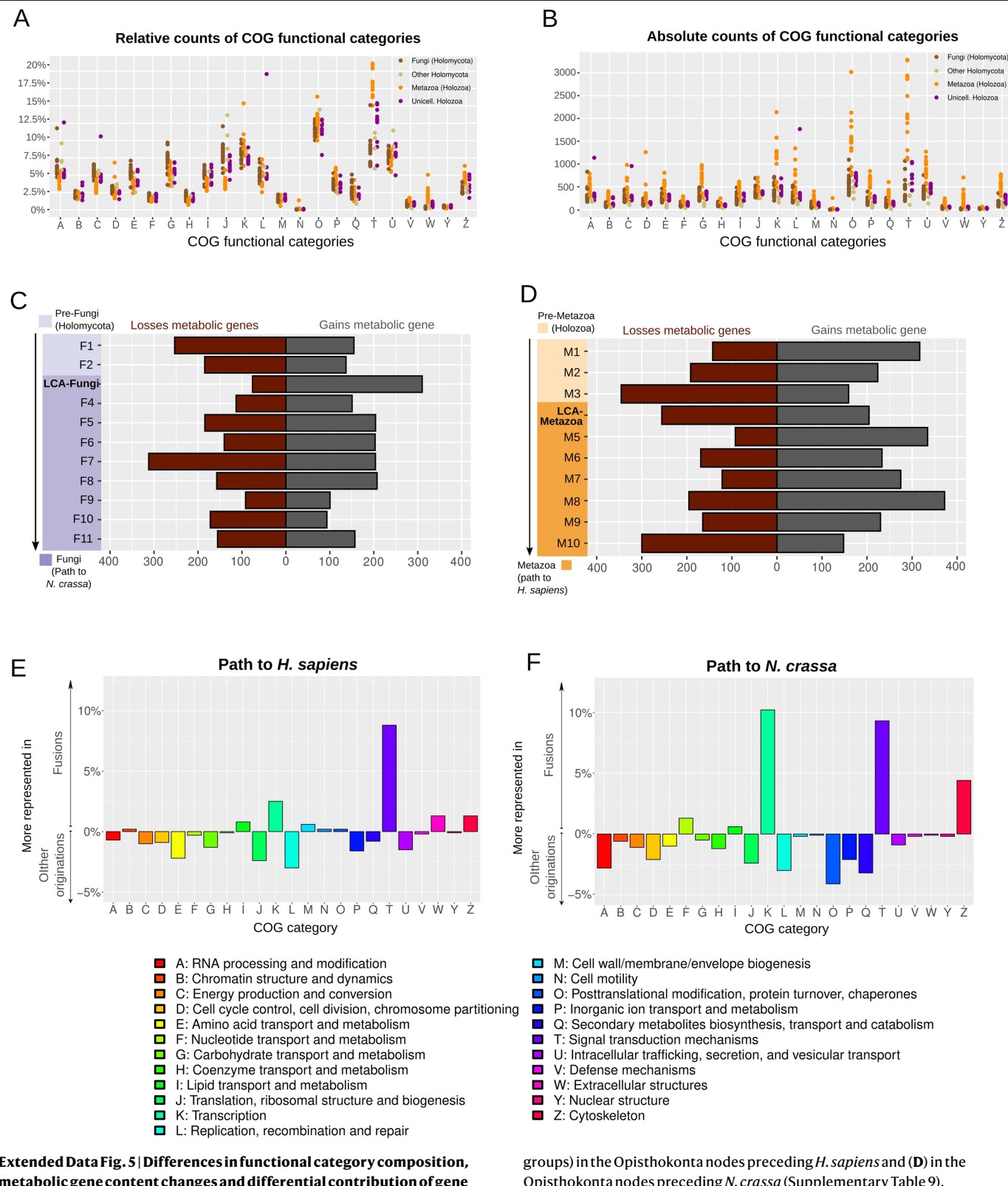

**Extended Data Fig. 5 | Differences in functional category composition, metabolic gene content changes and differential contribution of gene fusion originations vs non-fusion originations to each functional category in Opisthokonta.** (**A**) Relative and (**B**) absolute counts of functional categories in the opisthokont species from euk_db (Supplementary Tables 7 and 8, respectively). (**C**) Gains and losses of metabolic genes (KEGG orthology groups) in the Opisthokonta nodes preceding *H. sapiens* and (**D**) in the Opisthokonta nodes preceding *N. crassa* (Supplementary Table 9). (**E**) Differential representation of functional categories among fusion originations vs non-fusion originations in the Opisthokonta nodes preceding *H. sapiens* and (**F**) in the Opisthokonta nodes preceding *N. crassa* (Supplementary Table 10).

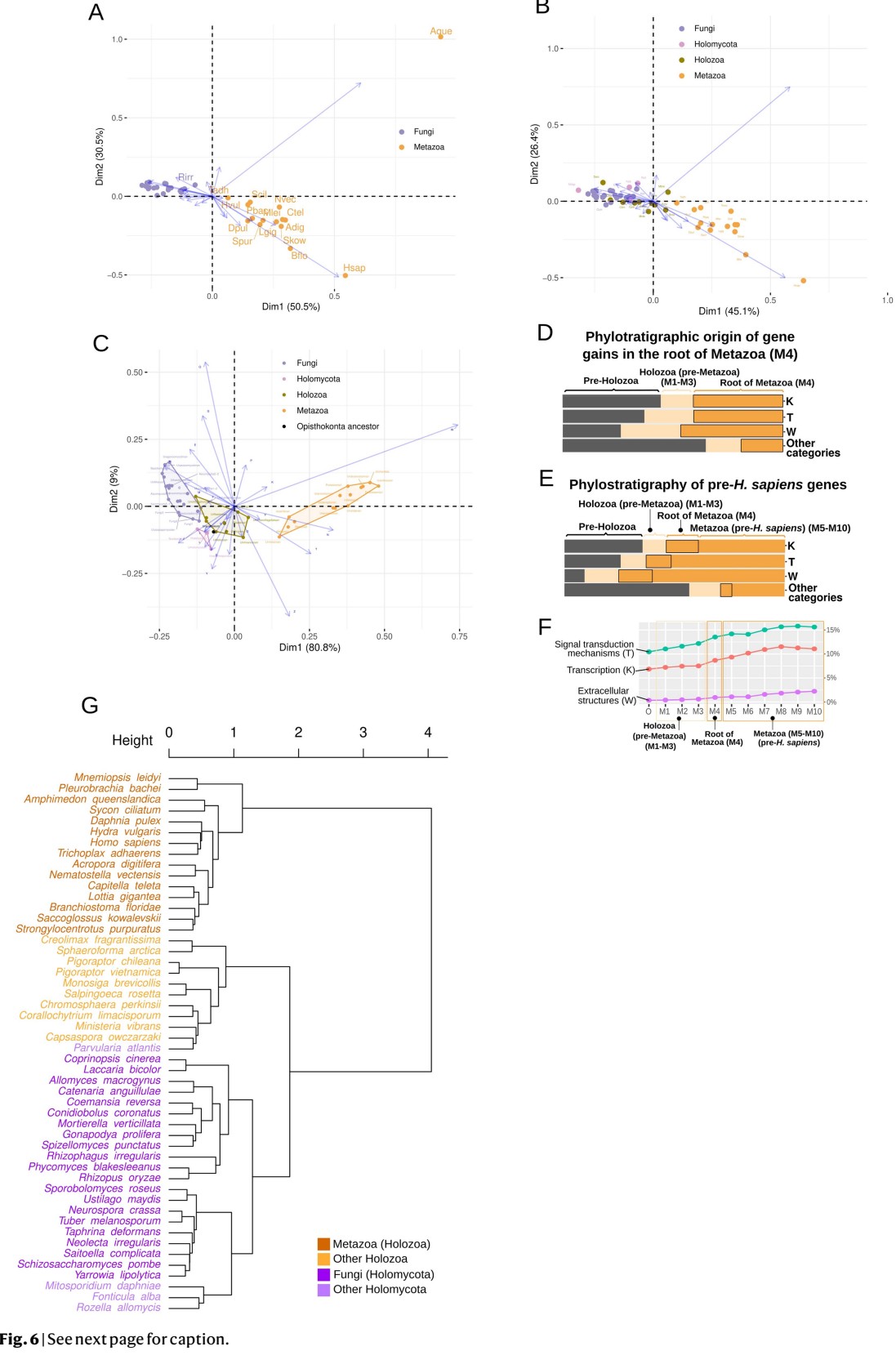

**Extended Data Fig. 6** | See next page for caption.

**Extended Data Fig. 6 | Correspondence analyses contribution biplots on functional category compositions in Opisthokonta, phylostratigraphic analyses of functional category changes in the evolutionary path towards extant Metazoa and clustering of Opisthokonta species based on gene family content composition.** Correspondence Analyses contribution biplot for the relative representation of functional categories (Supplementary Table 7) in the species from euk_db dataset representing (**A**) Metazoa and Fungi (**B**) Opisthokonta (i.e., Metazoa, Fungi, and also the other Holozoa and Holomycota sampled, Supplementary Table 4), and (**C**) every ancestor represented by an internal node in the Opisthokonta phylogeny (see Fig. 7 in Supplementary Information 3 for a mapping of every ancestral lineage to the phylogeny). (**D**) Phylostratigraphic origin of each functional category for those gene families that experienced increments in copy number (either gene gains or gene originations) in the last common ancestor of Metazoa for each functional category (Supplementary Table 12). (**E**) Phylostratigraphy of the ancestral gene content of *Homo sapiens* for each functional category (Supplementary Table 11). (**F**) Increment in the relative representation of functional categories which are particularly important for animal multicellularity since the divergence of Opisthokonta (Supplementary Table 13). (**G**) Similarities in gene family (orthogroups) composition between all the Opisthokonta species included in our study. We first computed the raw similarity value for each pair of species by inspecting those gene families found in both species and adding up for each of these families the lowest copy number value found among the two species. Each raw similarity value was then normalized by multiplying it by two and dividing it by the maximum possible similarity value that could have been found for that pair of species, which corresponds to the sum of members that every gene family has in the two species (species-specific families were not considered) (Supplementary Table 14). The dendrogram was reconstructed using the 'ward.D' method from the R package hclust.

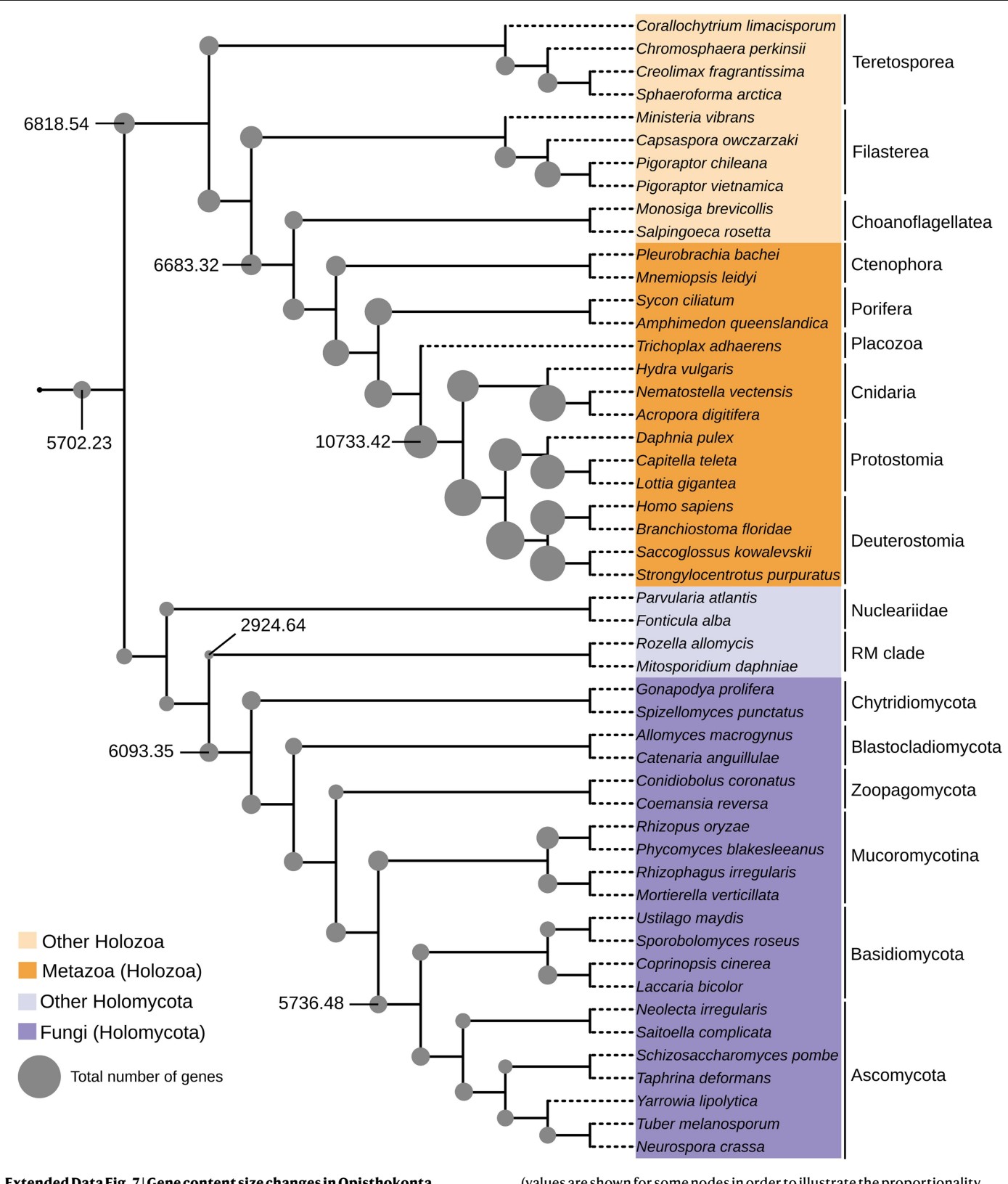

**Extended Data Fig. 7 | Gene content size changes in Opisthokonta evolution.** Gene content size inferred for every ancestral node of the Opisthokonta phylogeny as shown by the size of corresponding pie chart (values are shown for some nodes in order to illustrate the proportionality between the diameter size and the numeric values).

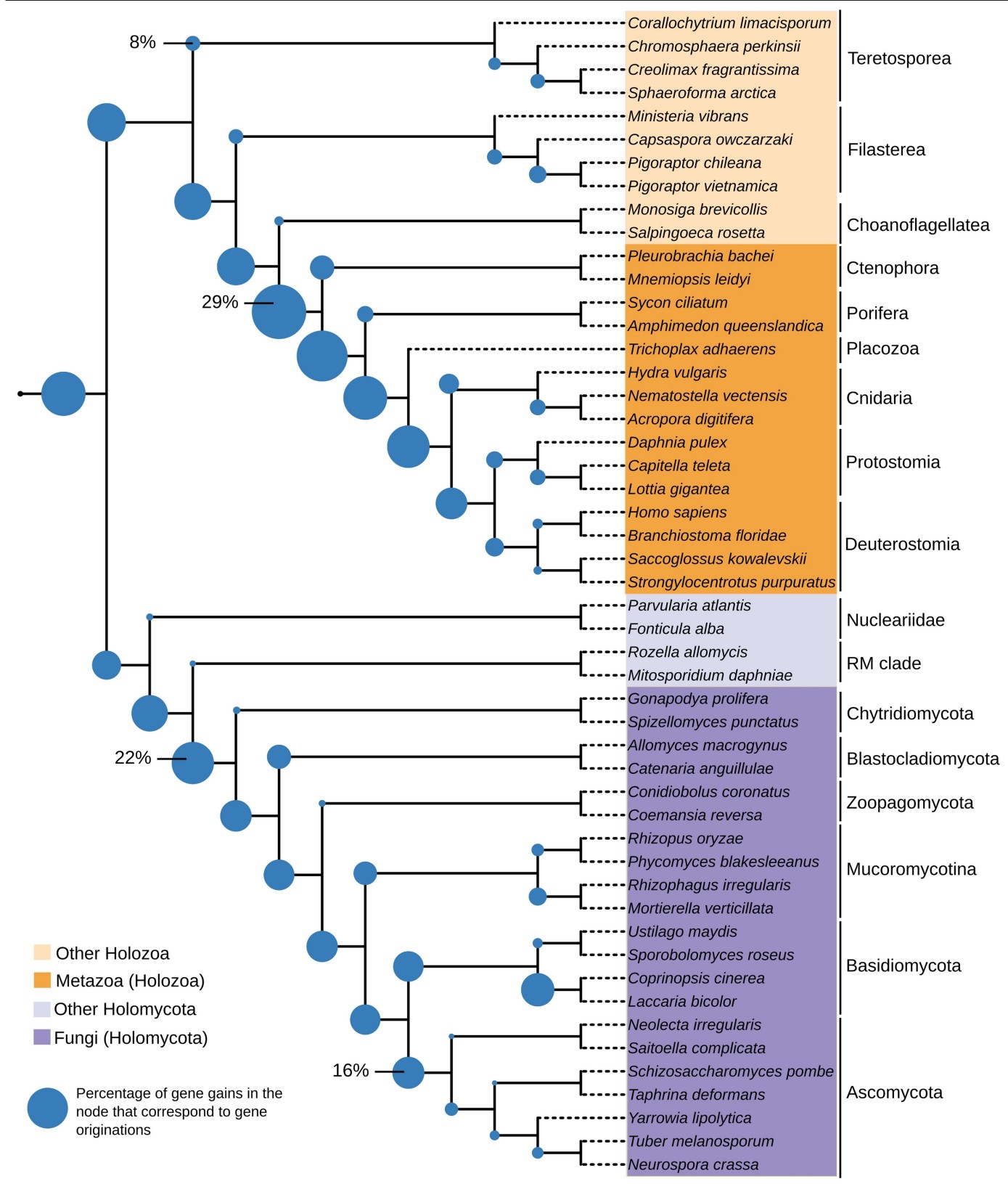

**Extended Data Fig. 8 | Relative contribution of gene originations to gene gains in Opisthokonta evolution.** Percentages of gene gains corresponding to gene originations (including gene fusions) inferred for every ancestral node of the Opisthokonta phylogeny as shown by the size of corresponding pie chart (values are shown for some nodes in order to illustrate the proportionality between the diameter size and the numeric values).

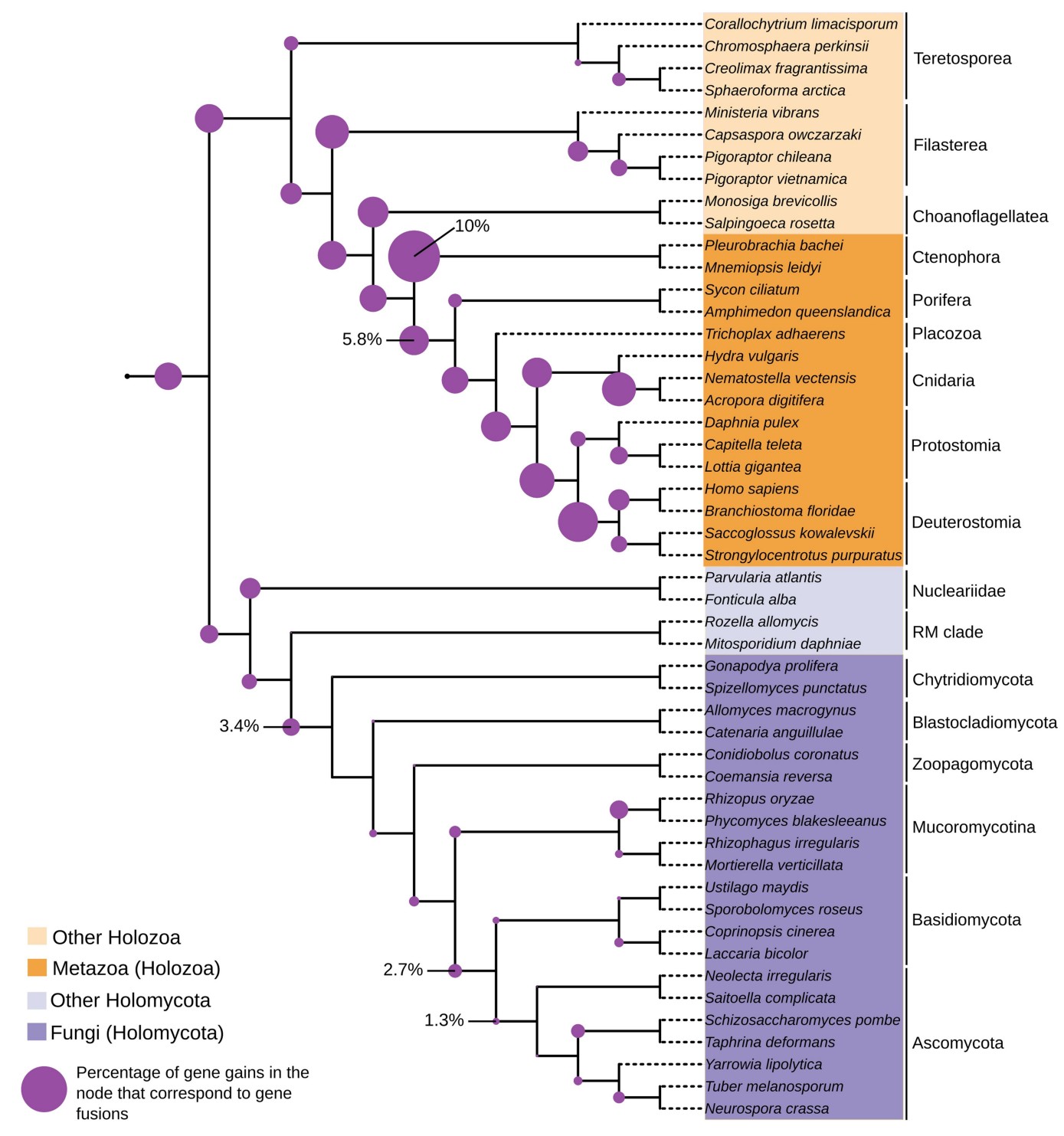

**Extended Data Fig. 9 | Relative contribution of gene fusions to gene gains in Opisthokonta evolution.** Percentages of gene gains corresponding to gene fusions inferred for every ancestral node of the Opisthokonta phylogeny as shown by the size of corresponding pie chart (values are shown for some nodes in order to illustrate the proportionality between the diameter size and the numeric values).

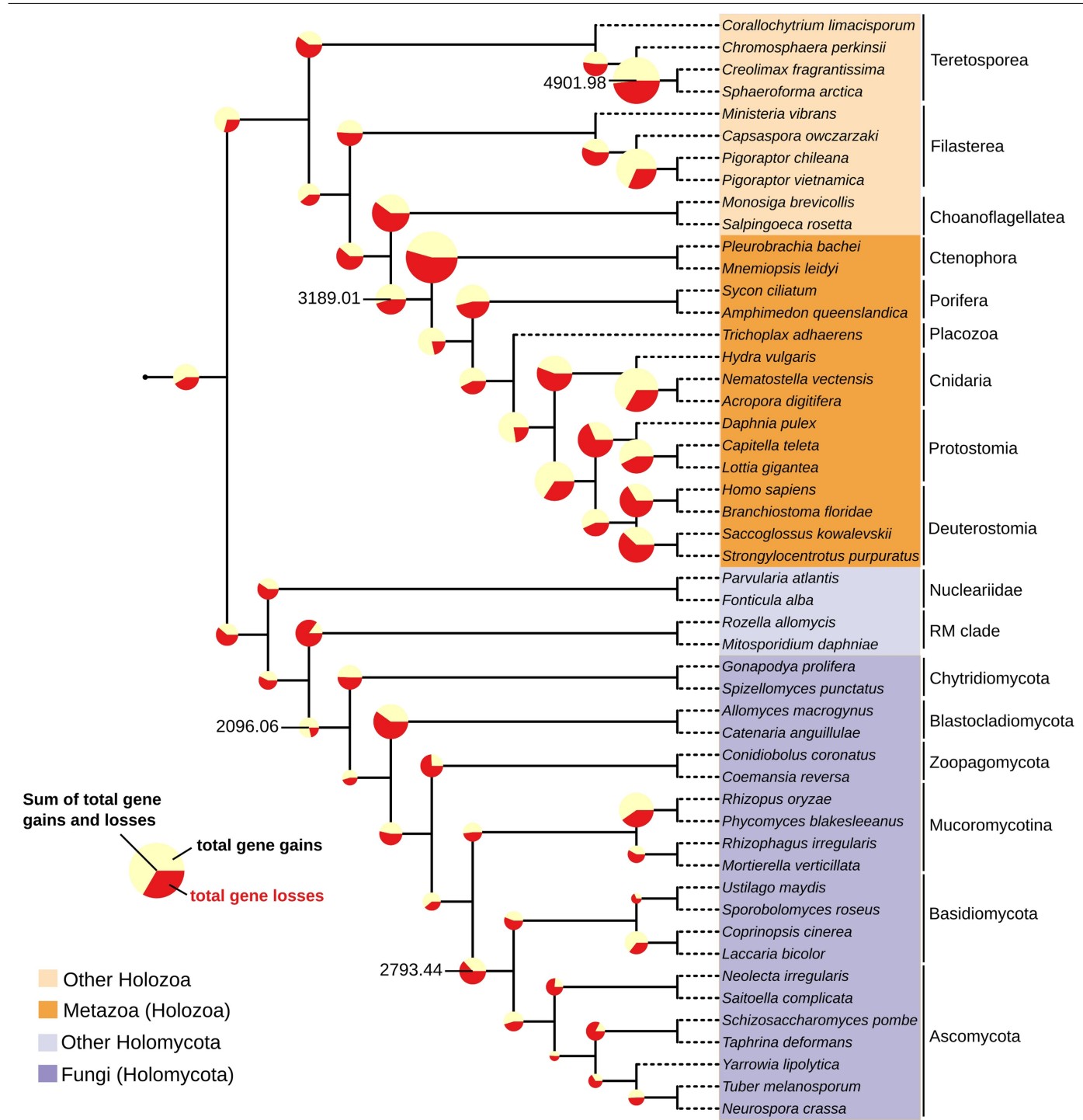

**Extended Data Fig. 10 | Gene gains and losses in Opisthokonta evolution.** Sum of gene gains and gene losses (and the fraction of the sum corresponding to each one) inferred for the internal nodes of the Opisthokonta phylogeny as shown by the size of corresponding pie chart (values are shown for some nodes in order to illustrate the proportionality between the diameter size and the numeric values).

# Reporting Summary

## Statistics

For all statistical analyses, confirm that the following items are present in the figure legend, table legend, main text, or Methods section.

| n/a | Confirmed | |
|---|---|---|
| ☐ | ☒ | The exact sample size (*n*) for each experimental group/condition, given as a discrete number and unit of measurement |
| ☒ | ☐ | A statement on whether measurements were taken from distinct samples or whether the same sample was measured repeatedly |
| ☐ | ☒ | The statistical test(s) used AND whether they are one- or two-sided<br>*Only common tests should be described solely by name; describe more complex techniques in the Methods section.* |
| ☒ | ☐ | A description of all covariates tested |
| ☒ | ☐ | A description of any assumptions or corrections, such as tests of normality and adjustment for multiple comparisons |
| ☐ | ☒ | A full description of the statistical parameters including central tendency (e.g. means) or other basic estimates (e.g. regression coefficient) AND variation (e.g. standard deviation) or associated estimates of uncertainty (e.g. confidence intervals) |
| ☐ | ☒ | For null hypothesis testing, the test statistic (e.g. *F*, *t*, *r*) with confidence intervals, effect sizes, degrees of freedom and *P* value noted<br>*Give P values as exact values whenever suitable.* |
| ☐ | ☒ | For Bayesian analysis, information on the choice of priors and Markov chain Monte Carlo settings |
| ☒ | ☐ | For hierarchical and complex designs, identification of the appropriate level for tests and full reporting of outcomes |
| ☒ | ☐ | Estimates of effect sizes (e.g. Cohen's *d*, Pearson's *r*), indicating how they were calculated |

*Our web collection on statistics for biologists contains articles on many of the points above.*

## Software and code

Policy information about availability of computer code

| Data collection | Genomic data was manually collected from online repositories, no software was used for this purpose. |
|---|---|
| Data analysis | Custom algorithms and scripts that are central to the research, all written in Python (3.6.4), are described in Material and methods and Supplementary Information. Only open source software was used for data analysis: BRAKER1 v1.9, PASA v2.0.2, BLASTp v2.5, OrthoFinder v2.7, MAFFT v7.123b, trimAl v1.4.rev15, IQ-TREE v1.6.7, PhyloBayes-MPI v1.8, DIAMOND v0.8.22.84, ALE v0.4, CompositeSearch v1.0, PfamScan.pl v1.0, eggNOG-mapper v1.0.3-3-g3e2272, trimmomatic v0.36, FastQC v0.11.5, nxtrim v0.4.1, SPAdes v3.10.1, GeneMark-ES/ET v4.33, AUGUSTUS v3.1.0, SEECER v0.1.3, TopHat v2.1.1, Databionics ESOM Tools v1.0, bowtie2 v2.2.9, CD-HIT v4.6, CONCOCT v0.4.1, IDBA-UD v1.1.1, Ray Meta v.2.3.1, BUSCO v1.22, QUAST v4.2, samtools 1.3.1, PASA v2.0.2, blat v35x1, GMAP v2015-12-31, RepeatMasker version open-4.0.6, RepeatModeler v1.0.4, TransDecoder.LongOrfs v3.0.1, Trinity v2.2.0, pandas v1.0.5, Keras v2.4.3, scikit-learn v0.23.1, R v3.6.3, numpy v1.18.5.<br><br>The custom code developed in this study is available at https://doi.org/10.5281/zenodo.6586559 |

For manuscripts utilizing custom algorithms or software that are central to the research but not yet described in published literature, software must be made available to editors and reviewers. We strongly encourage code deposition in a community repository (e.g. GitHub). See the Nature Portfolio guidelines for submitting code & software for further information.

## Data

Policy information about availability of data

All manuscripts must include a data availability statement. This statement should provide the following information, where applicable:

- Accession codes, unique identifiers, or web links for publicly available datasets
- A description of any restrictions on data availability
- For clinical datasets or third party data, please ensure that the statement adheres to our policy

The raw sequence data and assembled genomes generated in this study have been deposited at the European Nucleotide Archive (project accession code PRJEB52884). The genome assemblies have also been deposited at Figshare (https://doi.org/10.6084/m9.figshare.19895962.v1). Protein sequences of the species used in this study were downloaded from the GenBank public databases (https://www.ncbi.nlm.nih.gov/protein/), Uniprot (https://www.uniprot.org/), JGI genome database (https://genome.jgi.doe.gov/portal/) and Ensembl genomes (https://www.ensembl.org). The following specific databases were also used in this study: Pfam A v29 (https://pfam.xfam.org/), EggNOG emapperdb-4.5.1 (http://eggnog5.embl.de) and UniProt reference proteomes Release 2016_02 (https://www.uniprot.org/). The supporting data files of this study are available in Figshare (https://doi.org/10.6084/m9.figshare.13140191.v1).

# Field-specific reporting

Please select the one below that is the best fit for your research. If you are not sure, read the appropriate sections before making your selection.

☐ Life sciences ☐ Behavioural & social sciences ☒ Ecological, evolutionary & environmental sciences

For a reference copy of the document with all sections, see nature.com/documents/nr-reporting-summary-flat.pdf

# Life sciences study design

All studies must disclose on these points even when the disclosure is negative.

| | |
|---|---|
| Sample size | *Describe how sample size was determined, detailing any statistical methods used to predetermine sample size OR if no sample-size calculation was performed, describe how sample sizes were chosen and provide a rationale for why these sample sizes are sufficient.* |
| Data exclusions | *Describe any data exclusions. If no data were excluded from the analyses, state so OR if data were excluded, describe the exclusions and the rationale behind them, indicating whether exclusion criteria were pre-established.* |
| Replication | *Describe the measures taken to verify the reproducibility of the experimental findings. If all attempts at replication were successful, confirm this OR if there are any findings that were not replicated or cannot be reproduced, note this and describe why.* |
| Randomization | *Describe how samples/organisms/participants were allocated into experimental groups. If allocation was not random, describe how covariates were controlled OR if this is not relevant to your study, explain why.* |
| Blinding | *Describe whether the investigators were blinded to group allocation during data collection and/or analysis. If blinding was not possible, describe why OR explain why blinding was not relevant to your study.* |

# Behavioural & social sciences study design

All studies must disclose on these points even when the disclosure is negative.

| | |
|---|---|
| Study description | *Briefly describe the study type including whether data are quantitative, qualitative, or mixed-methods (e.g. qualitative cross-sectional, quantitative experimental, mixed-methods case study).* |
| Research sample | *State the research sample (e.g. Harvard university undergraduates, villagers in rural India) and provide relevant demographic information (e.g. age, sex) and indicate whether the sample is representative. Provide a rationale for the study sample chosen. For studies involving existing datasets, please describe the dataset and source.* |
| Sampling strategy | *Describe the sampling procedure (e.g. random, snowball, stratified, convenience). Describe the statistical methods that were used to predetermine sample size OR if no sample-size calculation was performed, describe how sample sizes were chosen and provide a rationale for why these sample sizes are sufficient. For qualitative data, please indicate whether data saturation was considered, and what criteria were used to decide that no further sampling was needed.* |
| Data collection | *Provide details about the data collection procedure, including the instruments or devices used to record the data (e.g. pen and paper, computer, eye tracker, video or audio equipment) whether anyone was present besides the participant(s) and the researcher, and whether the researcher was blind to experimental condition and/or the study hypothesis during data collection.* |
| Timing | *Indicate the start and stop dates of data collection. If there is a gap between collection periods, state the dates for each sample cohort.* |
| Data exclusions | *If no data were excluded from the analyses, state so OR if data were excluded, provide the exact number of exclusions and the* |

| Data exclusions | *rationale behind them, indicating whether exclusion criteria were pre-established.* |
| Non-participation | *State how many participants dropped out/declined participation and the reason(s) given OR provide response rate OR state that no participants dropped out/declined participation.* |
| Randomization | *If participants were not allocated into experimental groups, state so OR describe how participants were allocated to groups, and if allocation was not random, describe how covariates were controlled.* |

# Ecological, evolutionary & environmental sciences study design

All studies must disclose on these points even when the disclosure is negative.

| Study description | We have reconstructed the tempo and mode of the genomic divergence process that accompanied the origin of animals and fungi since the divergence of the last shared common ancestor of both groups (Opisthokonta). For it, we first analysed in a comparative manner the gene contents of modern animals and fungi in order the identify the fundamental genomic differences between them. Then, based on the gene content of extant representatives of both groups as well as from other opisthokont lineages that branch between Metazoa and Fungi, we used a phylogenetic approach (a gene-tree/species-tree reconciliation software) to reconstruct the ancestral gene contents and the genetic turnover occurred at every ancestral lineage in the Opisthokonta phylogeny. This methodological workflow allowed us to compare the genetic changes that occurred in the evolutionary path towards both groups at the level of (i) gene gains and losses, (ii) functional specialization of the gene content, and (iii) between-group differences in the relative preference for a series of mechanisms that can operate as sources of new genes. |
| Research sample | This study re-analyses mostly publicly available protein sequence predictions of species gene contents (hereafter referred to as gene content) from genomic or transcriptomic data, as well as the gene contents annotated from four newly sequenced protist opisthokont species in this mansucript: Parvularia atlantis, Ministeria vibrans, Pigoraptor vietnamica and Pigoraptor chileana. In particular, the dataset included the gene contents from 16 metazoans (Holozoa, Opisthokonta), 18 non-metazoan Holozoa, 21 Fungi (Holomycota, Opisthokonta), 4 non-fungal Holomycota, and 24 non-opisthokont eukaryotes. The species included in the dataset were chosen in order to maximize the taxonomic representation of the distinct metazoan and fungal groups, while also from the distinct opisthokont lineages that branch between animals and fungi. Gene content data from other eukaryotic groups were also included in order to ensure that the reconstructed gene trees could include phylogenetic information also from the evolutionary history preceding the origin of the specific group of interest for this study (Opstihokonta). Sequences from prokaryotes and virsuses were also added into the dataset for this same purpose, although they were incorporated after the clustering of the sampled eukaryotic sequences into orthogroups (see Methods section in the manuscript for a detailed explanation). Prokaryotic and viral sequences were incorporated from a database that included 8231104, 331476 and 20955 sequences from Bacteria, Archaea and viruses, respetively, which correspond to all Uniprot reference proteomes from these groups (release 2016_02). The final dataset (euk_db, see Methods section in the manuscript for details) that was used for the ancestral reconstruction analysis of gene content consisted of 1117614 eukaryotic sequences and 58017 non-eukaryotic sequences. |
| Sampling strategy | Sampling strategy: The dataset size was constrainted by the availability of genomic data and by some computational analyses. In particular, there were some methodological steps that were more computationally demanding than others, and for which a reduced version (euk_db) of the original dataset (draft_euk_db) was used (see Supplementary Table 4 for a description of the eukaryotic groups included in each dataset, and Methods section for an explanation of which dataset was used in every methodological step). The draft_euk_db dataset included gene content data from 83 eukaryotic species that represent all major eukaryotic groups. According to the interest of this study, the taxonomic representation of the eukaryotic supergroup Opisthokonta was prioritized (59 species). Only 9 opisthokont species were not included in the reduced euk_db dataset, 8 of them because gene content annotations came from transcriptomic data, and one species (Oscarella carmela) because its reduced gene content size was suggestive of incomplete genomic data or of this being an outlier species with a highly reduced genome. A total of 15 species were from draft_euk_db were not included in euk_db. <br><br> The size of the taxon sampling used in this study (83 genomes) is in the same scale than the taxon samplings used in similar studies (e.g., 69 genomes in https://doi.org/10.7554/eLife.26036.005, 72 genomes in https://doi.org/10.1038/s41467-019-12085-w), with the advantage that we have incorporated four novel species genomes that we sequenced for two taxonomic groups that were previously represented by only one species genome each (Filasterea and Nucleariidea). This allowed us to use the most updated possible taxon sampling at genome level with regard to the taxonomic groups that are phylogenetically related to animals and to fungi (among which Filasterea and Nucleariidea are included, see Fig. 1). In phylogenetic analyses, statistical robustness is estimated using bootstrap support values or, in Bayesian analyses, posterior probabilities; low support values might indicate a lack of statistical power to distinguish between hypotheses of relationship. The uncertainty associated with our estimates is provided using these standard metrics. The high support values in the reconstructed phylogenies (see Supplementary Information 3-Fig. 1A,B) indicate that sampling was sufficient. |
| Data collection | The first author of the manuscript downloaded the genomic data from public genomic repositories (see Data availability statement). Data downloading procedure was recorded in an Excel file and is shown in Supplementary Table 4. |
| Timing and spatial scale | The genomic data were downloaded from 2016 to 2018 from the public genomic repositories as indicated in the 'Data collection' section, and more into detail in the Methods section. |
| Data exclusions | Transcriptomic data was only used for species tree reconstruction, in particular for those taxa with phylogenetically relevant positions in the context of the Opisthoknota phylogeny but for which genomic data is not yet available. Transcriptomic data was not used in the ancestral gene content reconstruction analyis because the gene contents that are obtained from transcriptome tend to be more inaccurate than those that are obtained when the genome sequence is also available. For example, uncollapsed transcriptome isoforms that may have been annotated as separate gene sequences can lead the reconciliation software to infer false gene duplications, unexpressed genes can be confused with gene losses, sequence contamination -which is harder to detect in |

| | transcriptomic data- can be confused with horizontal gene transfers, and partial fragments can lead to artifactual protein domain architectures which can confuse the algorithm used to detect gene fusion events. |
|---|---|
| Reproducibility | The raw genomic data, software, software versions and software parameters are specified in the Methods section, to enable all analyses to be reproduced or built upon as needed. Note that all analyses are done with bioinformatic methods and hence the reproducibility of the results is guaranteed as long as the same data, software versions and software parameters are used. |
| Randomization | The paper reports phylogenetic and comparative genomic analyses. As is standard for the field, the robustness of inferences was assessed using bootstrap support values and Bayesian posterior probabilities. |
| Blinding | Blinding was not relevant to the study design, because this was a phylogenetic and comparative genomic analysis of all of the available data (that is, the experimental design did not involve allocating data to groups). |

Did the study involve field work?  ☐ Yes  ☒ No

## Field work, collection and transport

| Field conditions | Describe the study conditions for field work, providing relevant parameters (e.g. temperature, rainfall). |
|---|---|
| Location | State the location of the sampling or experiment, providing relevant parameters (e.g. latitude and longitude, elevation, water depth). |
| Access & import/export | Describe the efforts you have made to access habitats and to collect and import/export your samples in a responsible manner and in compliance with local, national and international laws, noting any permits that were obtained (give the name of the issuing authority, the date of issue, and any identifying information). |
| Disturbance | Describe any disturbance caused by the study and how it was minimized. |

# Reporting for specific materials, systems and methods

We require information from authors about some types of materials, experimental systems and methods used in many studies. Here, indicate whether each material, system or method listed is relevant to your study. If you are not sure if a list item applies to your research, read the appropriate section before selecting a response.

### Materials & experimental systems

| n/a | Involved in the study |
|---|---|
| ☒ | ☐ Antibodies |
| ☐ | ☒ Eukaryotic cell lines |
| ☒ | ☐ Palaeontology and archaeology |
| ☒ | ☐ Animals and other organisms |
| ☒ | ☐ Human research participants |
| ☒ | ☐ Clinical data |
| ☒ | ☐ Dual use research of concern |

### Methods

| n/a | Involved in the study |
|---|---|
| ☒ | ☐ ChIP-seq |
| ☒ | ☐ Flow cytometry |
| ☒ | ☐ MRI-based neuroimaging |

## Antibodies

| Antibodies used | Describe all antibodies used in the study; as applicable, provide supplier name, catalog number, clone name, and lot number. |
|---|---|
| Validation | Describe the validation of each primary antibody for the species and application, noting any validation statements on the manufacturer's website, relevant citations, antibody profiles in online databases, or data provided in the manuscript. |

## Eukaryotic cell lines

Policy information about cell lines

| Cell line source(s) | Ministeria vibrans' culture was bought in ATCC (Ministeria vibrans Tong. ATCC 50519). Parvularia atlantis' (formerly Nuclearia sp.) culture was bought in ATCC (Nuclearia sp. ATCC 50694). The cultures of Pigoraptor vietnamica (formerly Opistho-1) and Pigoraptor chileana (formerly Opistho-2) descend from the environmental isolates (P. vietnamica from a Freshwater Lake, Vietnam, and P. chileana from freshwater temporary water body, Chile) used in the manuscript https://doi.org/10.1016/j.cub.2017.06.006 (see 'METHOD DETAILS' section). |
|---|---|
| Authentication | The presence of our organisms of interest in the sequenced cell lines was validated through microscopy observation and genetic analyses before genomic sequencing. The genes from the organisms of interests are found in the sequenced data, confirming their presence in the sequenced cell lines. |
| Mycoplasma contamination | The culture lines used in this study are not pure cell lines, and hence include other species besides our organisms of interest. For this reason, the genomic data produced was fully decontaminated by means of a comprehensive bioinformatic analyses consisting of distinct iterative rounds of decontamination, as described in Supplementary Information. |

| Commonly misidentified lines (See ICLAC register) | Name any commonly misidentified cell lines used in the study and provide a rationale for their use. |

# Palaeontology and Archaeology

| Specimen provenance | Provide provenance information for specimens and describe permits that were obtained for the work (including the name of the issuing authority, the date of issue, and any identifying information). Permits should encompass collection and, where applicable, export. |

| Specimen deposition | Indicate where the specimens have been deposited to permit free access by other researchers. |

| Dating methods | If new dates are provided, describe how they were obtained (e.g. collection, storage, sample pretreatment and measurement), where they were obtained (i.e. lab name), the calibration program and the protocol for quality assurance OR state that no new dates are provided. |

☐ Tick this box to confirm that the raw and calibrated dates are available in the paper or in Supplementary Information.

| Ethics oversight | Identify the organization(s) that approved or provided guidance on the study protocol, OR state that no ethical approval or guidance was required and explain why not. |

Note that full information on the approval of the study protocol must also be provided in the manuscript.

# Animals and other organisms

Policy information about studies involving animals; ARRIVE guidelines recommended for reporting animal research

| Laboratory animals | For laboratory animals, report species, strain, sex and age OR state that the study did not involve laboratory animals. |

| Wild animals | Provide details on animals observed in or captured in the field; report species, sex and age where possible. Describe how animals were caught and transported and what happened to captive animals after the study (if killed, explain why and describe method; if released, say where and when) OR state that the study did not involve wild animals. |

| Field-collected samples | For laboratory work with field-collected samples, describe all relevant parameters such as housing, maintenance, temperature, photoperiod and end-of-experiment protocol OR state that the study did not involve samples collected from the field. |

| Ethics oversight | Identify the organization(s) that approved or provided guidance on the study protocol, OR state that no ethical approval or guidance was required and explain why not. |

Note that full information on the approval of the study protocol must also be provided in the manuscript.

# Human research participants

Policy information about studies involving human research participants

| Population characteristics | Describe the covariate-relevant population characteristics of the human research participants (e.g. age, gender, genotypic information, past and current diagnosis and treatment categories). If you filled out the behavioural & social sciences study design questions and have nothing to add here, write "See above." |

| Recruitment | Describe how participants were recruited. Outline any potential self-selection bias or other biases that may be present and how these are likely to impact results. |

| Ethics oversight | Identify the organization(s) that approved the study protocol. |

Note that full information on the approval of the study protocol must also be provided in the manuscript.

# Clinical data

Policy information about clinical studies

All manuscripts should comply with the ICMJE guidelines for publication of clinical research and a completed CONSORT checklist must be included with all submissions.

| Clinical trial registration | Provide the trial registration number from ClinicalTrials.gov or an equivalent agency. |

| Study protocol | Note where the full trial protocol can be accessed OR if not available, explain why. |

| Data collection | Describe the settings and locales of data collection, noting the time periods of recruitment and data collection. |

| Outcomes | Describe how you pre-defined primary and secondary outcome measures and how you assessed these measures. |

# Dual use research of concern

Policy information about dual use research of concern

## Hazards

Could the accidental, deliberate or reckless misuse of agents or technologies generated in the work, or the application of information presented in the manuscript, pose a threat to:

No | Yes

☐ ☐ Public health

☐ ☐ National security

☐ ☐ Crops and/or livestock

☐ ☐ Ecosystems

☐ ☐ Any other significant area

## Experiments of concern

Does the work involve any of these experiments of concern:

No | Yes

☐ ☐ Demonstrate how to render a vaccine ineffective

☐ ☐ Confer resistance to therapeutically useful antibiotics or antiviral agents

☐ ☐ Enhance the virulence of a pathogen or render a nonpathogen virulent

☐ ☐ Increase transmissibility of a pathogen

☐ ☐ Alter the host range of a pathogen

☐ ☐ Enable evasion of diagnostic/detection modalities

☐ ☐ Enable the weaponization of a biological agent or toxin

☐ ☐ Any other potentially harmful combination of experiments and agents

# ChIP-seq

## Data deposition

☐ Confirm that both raw and final processed data have been deposited in a public database such as GEO.

☐ Confirm that you have deposited or provided access to graph files (e.g. BED files) for the called peaks.

**Data access links**
*May remain private before publication.*

> For "Initial submission" or "Revised version" documents, provide reviewer access links. For your "Final submission" document, provide a link to the deposited data.

**Files in database submission**

> Provide a list of all files available in the database submission.

**Genome browser session**
(e.g. UCSC)

> Provide a link to an anonymized genome browser session for "Initial submission" and "Revised version" documents only, to enable peer review. Write "no longer applicable" for "Final submission" documents.

## Methodology

**Replicates**

> Describe the experimental replicates, specifying number, type and replicate agreement.

**Sequencing depth**

> Describe the sequencing depth for each experiment, providing the total number of reads, uniquely mapped reads, length of reads and whether they were paired- or single-end.

**Antibodies**

> Describe the antibodies used for the ChIP-seq experiments; as applicable, provide supplier name, catalog number, clone name, and lot number.

**Peak calling parameters**

> Specify the command line program and parameters used for read mapping and peak calling, including the ChIP, control and index files used.

**Data quality**

> Describe the methods used to ensure data quality in full detail, including how many peaks are at FDR 5% and above 5-fold enrichment.

**Software**

> Describe the software used to collect and analyze the ChIP-seq data. For custom code that has been deposited into a community repository, provide accession details.

# Flow Cytometry

## Plots

Confirm that:

☒ The axis labels state the marker and fluorochrome used (e.g. CD4-FITC).

☒ The axis scales are clearly visible. Include numbers along axes only for bottom left plot of group (a 'group' is an analysis of identical markers).

☒ All plots are contour plots with outliers or pseudocolor plots.

☒ A numerical value for number of cells or percentage (with statistics) is provided.

## Methodology

| | |
|---|---|
| Sample preparation | Sorted cells were sampled from polyxenic protist cultures including the eukaryotic species of interest as well as an uncertain fraction of bacterial contamination |
| Instrument | BD FACSAria II cell sorter (Becton Dickinson, San Jose, CA). Model number: P5X10001 |
| Software | Facsdiva Software Version 6.1.2 |
| Cell population abundance | The final population sorted represented less than 1% of the total cells in the sample. The aim was to enrich the population of eukaryotic cells and to minimize the fraction of contaminantion in the sequenced metagenomic data. As expected, some contamination remained in the sequenced pool of sorted cells which was subsequently eliminated with a comprehensive bioinformatic pipeline that is thoroughtly explained in Supplementary Information 1. |
| Gating strategy | We used Forward Scatter (FSC) and Side Scatter (SCC) lasers together with the green fluorescence (FITC channel 525/50 nm) to target larger and complex eukaryotic cells that incorporated Lysotracker-green DND-26, which is eukaryotic specific. Next, we could discriminate which eukaryotic cells had a larger fraction of bacterial cells attached with the dye 5-Cyano-2,3-ditolyl tetrazolium chloride (CTC, PerCPCy5.5 channel 685/35 nm). We sorted the population of eukaryotic cells that presented the lowest CTC signal in order to minimize the fraction of bacterial contamination in the sorted cells. |

☒ Tick this box to confirm that a figure exemplifying the gating strategy is provided in the Supplementary Information.

# Magnetic resonance imaging

## Experimental design

| | |
|---|---|
| Design type | *Indicate task or resting state; event-related or block design.* |
| Design specifications | *Specify the number of blocks, trials or experimental units per session and/or subject, and specify the length of each trial or block (if trials are blocked) and interval between trials.* |
| Behavioral performance measures | *State number and/or type of variables recorded (e.g. correct button press, response time) and what statistics were used to establish that the subjects were performing the task as expected (e.g. mean, range, and/or standard deviation across subjects).* |

## Acquisition

| | |
|---|---|
| Imaging type(s) | *Specify: functional, structural, diffusion, perfusion.* |
| Field strength | *Specify in Tesla* |
| Sequence & imaging parameters | *Specify the pulse sequence type (gradient echo, spin echo, etc.), imaging type (EPI, spiral, etc.), field of view, matrix size, slice thickness, orientation and TE/TR/flip angle.* |
| Area of acquisition | *State whether a whole brain scan was used OR define the area of acquisition, describing how the region was determined.* |

Diffusion MRI ☐ Used ☐ Not used

## Preprocessing

| | |
|---|---|
| Preprocessing software | *Provide detail on software version and revision number and on specific parameters (model/functions, brain extraction, segmentation, smoothing kernel size, etc.).* |
| Normalization | *If data were normalized/standardized, describe the approach(es): specify linear or non-linear and define image types used for transformation OR indicate that data were not normalized and explain rationale for lack of normalization.* |
| Normalization template | *Describe the template used for normalization/transformation, specifying subject space or group standardized space (e.g.* |

| Normalization template | *original Talairach, MNI305, ICBM152) OR indicate that the data were not normalized.* |
|---|---|

| Noise and artifact removal | *Describe your procedure(s) for artifact and structured noise removal, specifying motion parameters, tissue signals and physiological signals (heart rate, respiration).* |
|---|---|

| Volume censoring | *Define your software and/or method and criteria for volume censoring, and state the extent of such censoring.* |
|---|---|

## Statistical modeling & inference

| Model type and settings | *Specify type (mass univariate, multivariate, RSA, predictive, etc.) and describe essential details of the model at the first and second levels (e.g. fixed, random or mixed effects; drift or auto-correlation).* |
|---|---|

| Effect(s) tested | *Define precise effect in terms of the task or stimulus conditions instead of psychological concepts and indicate whether ANOVA or factorial designs were used.* |
|---|---|

Specify type of analysis: ☐ Whole brain ☐ ROI-based ☐ Both

| Statistic type for inference<br>(See Eklund et al. 2016) | *Specify voxel-wise or cluster-wise and report all relevant parameters for cluster-wise methods.* |
|---|---|

| Correction | *Describe the type of correction and how it is obtained for multiple comparisons (e.g. FWE, FDR, permutation or Monte Carlo).* |
|---|---|

## Models & analysis

| n/a | Involved in the study |
|---|---|
| ☐ ☐ | Functional and/or effective connectivity |
| ☐ ☐ | Graph analysis |
| ☐ ☐ | Multivariate modeling or predictive analysis |

| Functional and/or effective connectivity | *Report the measures of dependence used and the model details (e.g. Pearson correlation, partial correlation, mutual information).* |
|---|---|

| Graph analysis | *Report the dependent variable and connectivity measure, specifying weighted graph or binarized graph, subject- or group-level, and the global and/or node summaries used (e.g. clustering coefficient, efficiency, etc.).* |
|---|---|

| Multivariate modeling and predictive analysis | *Specify independent variables, features extraction and dimension reduction, model, training and evaluation metrics.* |
|---|---|

