## [Peer Review File · Nature]

Manuscript Title: Divergent genomic trajectories predate the origin of animals and fungi

Reviewer Comments & Author Rebuttals

Reviewer Reports on the Initial Version:

Referees' comments:

Referee #1 (Remarks to the Author):

I got this paper and really wanted to like it a lot. I feel that there is almost no methodological problem with the paper - though as you can see, I have a question about what might be circular reasoning in one part. My overall impression is of an interesting paper, but I don't feel particularly surprised by the results. Fungi are metabolic geniuses - this is not surprising, while animals have remodelled their genes extensively, through rounds of duplications, lots of recombination etc. The paper is definitely interesting, but perhaps with a bit of clarifying the small number of points I have to make, it would be easier for me to see how much of a surprising set of data we have here. Personally, I would like to see it published - the thinking is definitely quite novel and the contribution of new key genomes has made this "transition" of animals/fungi very interesting indeed.

I am somewhat at a loss for figure 1 (and its associated discussion) why there are approximately 28 internal branches that are not labeled at all. The authors seem to have picked the branches they wished to label. The ladderizing of the tree is just an arbitrary way of viewing it. Why did the authors focus on approximately 50% of the internal branches and ignore the rest?

Also for figure 1, could the authors say why they do not represent branch lengths on their tree, only topology?

The same can be said for figure 2 - why are branch lengths not represented on the tree? If it is just aesthetics, then this is probably not a good enough reason.

From lines 133 to 149, I am struggling to see how this is not circular reasoning. It seems to me that both the ML classifier and the correspondance analysis- based approach are both identifying fungal- and animal- enriched patterns and then asking whether these patterns are seen in the data. While the fungal-likeness is seen in the unicellular ancestors of animals, there was always going to be a particular branch when it flipped over - it had to flip over at some point. Again, if the tree in figure 1A had branch lengths on it, it would be easier to evaluate the significance of this flip-over.

The infographic of figure 4 didn't provide a great deal of enlightenment and I found it difficult to read.

The paper reports some key genomes, the statistics are appropriately used and implemented

(notwithstanding my question about the circularity of reasoning), the conclusions are robust and the authors do not "oversell" the results and the paper is clearly written.

Referee #2 (Remarks to the Author):

Ocana-Pallares et al.

Divergent genomic trajectories predate the origin of animals and fungi

The authors present a remarkably multifaceted genomics-informed investigation of the evolution of Metazoa and Fungi. These two lineages comprise the best known members of the two main groups of Opisthokonts (animals, fungi and various protists). Focusing on genomes from novel protist lineages that branch 'between' animals and fungi, they provide a window on the patterns and processes that contributed to the emergence these two groups. Some of their conclusions are somewhat confirmatory relative to previous ideas (e.g., role of gene duplication in early animal evolution, more lateral gene transfer in fungi and their protist progenitors relative to on the animal side). But I have never seen this question addressed so comprehensively and definitively, in such a way that makes it so clear what transpired (genetically speaking) on the deepest branches leading to extant Metazoa and Fungi -- and how the seeds were set in so very different ways on the two sides of the Opisthokonta. The really novel parts are where the authors dive into the changes in 'gene functional categories' and '% metabolic genes'. I should think this paper will be very well received and well read.

I also must tip my hat to the authors for presenting an understandably abbreviated version of their truly vast set of results in a way that can be followed. Unlike most papers done at this scale, I found this one very accessible and relatively straightforward to go back and forth between the main text and the supp info (figs and accompanying text). It's not perfect (see below), but a lot of work clearly went into the structure and content of the main text figures.

Specific Comments

-The nature of the new genomic data isn't clear enough. It's obviously novel and very important, but what kind of data is it, exactly? Assembled genomes (and presumably transcriptomes)? The abstract says 'four novel genomes', but nowhere else is this clearly addressed and diving into the details in the large amount of supp data / info, it's clear that it's more metagenomic in some cases since these new protists are very difficult to work with. It is understandable that the authors can't easily summarize the nature of the data they acquired and analyzed. But nowhere (that I could find) is there a single table that summarizes what was obtained for each of the four novel protists (it seems to be spread across at least 4 SI Tables). People in the field will want this. They will want to be able to access this information and integrate the new data into their analyses, and assess, e.g., the extent of partiality of the new data (and what they will and will not be able to do with it). I don't question the rigor of their data inclusion / exclusion processes (ESOM maps, etc.). But the authors would do well to provide a single SI table that puts the most important bits all in one place.

-Abstract: "...different preferences for sources of gene gains". Change to 'gene gain/ evolution processes' or something similar? Makes the reader (me at least) think of LGT, which only appears to have been a significant factor on the fungal side.

-Abstract: "We show that animals took advantage of a tendency of gaining genes of multicellularity-related categories which started in the pre-metazoan ancestors."

Using the term 'multicellularity-related category' made me pause because it sort of sounds as though it's a category that existed prior to multicellularity. I don't have a clean suggestion for a rephrase, but something along the lines of:

"We show that pre-metazoan ancestors took advantage of a tendency of gaining genes that today are associated with multicellularity".

Additional comments

Line 73: Filasterea (Holozoa) is the closest KNOWN relative group to animals...
Also, 'relative group' doesn't sound quite right.

Line 140: this should be tempo and modes, because there doesn't have to be (and there doesn't appear to be) just one mode.

Line 306: "Interestingly, these same categories underwent losses in the pre-fungal ancestors (Fig. 1B), situating the immediate ancestors of Fungi and Metazoa in completely different latent potentials from a genomic perspective."

I think 'completely different' is too strong here. Rather different? Substantially different?

Referee #3 (Remarks to the Author):

Review Ocaña-Pallarès et al. 2022

This paper represents a very large-scale analysis investigating the genome evolution between animals and fungi. It is well written and easy to follow (although improvements could be made), it is an interesting read. The methods used by the authors seem strong and repeatable. The supplementary are exhaustive and really detailed. The release of these four new genomes is an important resource for the community. The paper itself does not include any new surprises. It just

brings together a lots of thoughts that have been simmering in the field in different sub-disciplines and perspectives. It therefore does justice to a large scale evaluation of these ideas. I think the manuscript is worth publishing and will be a very important reference for the field.

Comment 1:

Is this study focusing only on genomic data? Are the 1,463,920 protein sequences from 83 eukaryotic species all coming from genomes? Did the authors avoid using proteomes coming from transcriptomic data? If so, it should be stated, clarified, and justified why.

I personally do not see a reason (even when transcriptome data can be limited) for not using proteomes coming transcriptomes (if required) and I cannot see a clearer example than using the only available transcriptome from the key group the aphelids (e.g. *Paraphelidium tribonemae*) for these analyses. Using this transcriptome would certainly provide a control for the trends they report at multiple nodes.

To this point it seems clear that aphelids are the sister lineage to Fungi (making Opisthosporidia paraphyletic) (Torruella et al. 2018; <https://doi.org/10.1038/s42003-018-0235-z> and Tikhonenkov et al. 2020; <https://doi.org/10.1016/j.cub.2020.08.061>). Thus, given their phylogenetic position is surprising that the authors make no mention of aphelids in the text. What is more, beyond their newly proposed phylogenetic position it has been shown that some hyphal multicellularity related categories (e.g. cell wall biogenesis/remodeling' and 'transcriptional regulation') evolved in the ancestor of aphelids and fungi (Galindo et al. 2021; <https://doi.org/10.1038/s41467-021-25308-w>). I understand that remaking all analysis for only one transcriptome can be troublesome, however, aphelids are in a key position to understand the picture the authors are trying to draw.

For example: Given your results the fungal root (F3) is the node with the biggest with the largest fraction of genes gains and metabolic expansion. However, part of this fraction could be explained by the non-present aphelid node.

The same could be said about *Olpidium bornovanus*, in this case there is both a genome and transcriptome which have been recently obtained (Chang et al. 2021; <https://doi.org/10.1038/s41598-021-82607-4>) showing that *Olpidium* falls within its on phylum which branches right at the phylogenetic border between zoosporic and non-zoosporic fungi. Adding this fungus to the analysis could have sharpen these results.

Comment 2:

Given its importance for multicellularity (at least in multicellular fruiting-body producing fungi): Is there any reason why cell adhesion, or adhesion-related genes are not explored (or at least not shown) in the functional categories? If adhesion is by any chance included within other functional category (e.g. cytoskeleton) It should be address on its own, since as seen in previous studies (Kiss et al. 2019; <https://doi.org/10.1038/s41467-019-12085-w> and Galindo et al. 2021; <https://doi.org/10.1038/s41467-021-25308-w> (Supplementary Fig 13)) adhesion-related genes expanded in complex multicellular fungi (mainly Dikarya), thus having a crucial role in the current

(fruiting body-producing) hyphal-based multicellularity.

Comment 3:

As mentioned in line 151 it is not surprising that the overall composition of the Fungi-category established here is more protist-like than the Metazoan-category, however, why this is happening is not explored enough.

This result fits within the fact that many of the Fungal phyla (in particular the early diverged lineages) are in fact unicellular organisms (Chytriomycota, Blastocladiomycota, Zoopagomycota). This is one of the main differences between Metazoan and Fungi, in Metazoa multicellularity was present in their root, and in fungi evolved latter. The presence of unicellular organisms within fungi needs to be highlighted even more since it justifies the observed patterns. The definition of fungi is to this day problematic, and many fungal clades could be argued to be protist clades (there is a continuity of zoosporic unicellular organisms from aphelelids and through chytrids, sanchytrids-blastoclads and Olpidium) (Richards et al. 2017; 10.1128/microbiolspec.FUNK-0044-2017). I think the point 'what is a fungus is important and should be considered from line 226-228

Comment 4:

It is interesting how the authors show that the biggest shifts towards Fungi-related functional categories occurred in the nodes F5 and F7, however it has not been discussed why. Loss of flagellum? Gain of aerial spores? Gain of hyphal traits?

F5 is the node in which fungi lost their flagellated life-stages (zoospores) and started to have septate hypha-like structures, and F7 is the shift towards Dikarya, in which "true" multicellular fruit-body producing fungi evolved. I think these large evolutionary innovations fit perfectly into their results. Please highlight it.

Comment 5

I am struggling with figure 1. I find the mini graphs impenetrable. Do you need MG6-M10. Given the war zone that is the early animal phylogeny do you need M5. Can these be in supplementary materials? Likewise do you need F5-F11. Then you could have less mini graphs, make them bigger and order the around the key nodes with the tree on its 'back'. I think that would be more approachable.

Minor comments

- Line 48 – define Fusions? I think you mean gene fusions
- Line 51 – why reduced HGT in metazoans. Weissman barrier? Is this all metazoans? Given the controversy of the animal HGT stories, please be specific. I think you can be and still fit it in an abstract.
- Line 74: Calling Fungi + "Opisthosporidia" the 'Fungi sensu latu' may be a big stretch, since Microsporidia and rozellids are far from the canonical definition of fungi. Please refer to

Nuclearia+“Opisthosporidia”+Aphelids+Fungi as Holomycota, and only refer to true fungi (Chytridiomycota and on) as fungi. Also refer to “Opisthosporidia” using quotes, given its paraphyletic nature.

- Overall, I am missing in the introduction a paragraph in which the authors define (as for them and as for this paper) what is fungi and what is Holomycota, since terms like (non-fungal opisthokonts; line 152-153) are terms that can create confusion.

- Line 84-89: It is incredibly interesting the super-high intron rate of Parvularia. I would highlight even more that Parvularia has by far the highest intron numbers of any eukaryote but at the same time its introns are one of the shortest. What are these introns? Any further exploration? Are they palindromic repetitive sequences? Short tandem repeats? Do they come from transposons? I think that it should definitely be explored (maybe in a follow up paper).

- We know Parvularia and Fonticula share the same clade within nucleariids (Galindo et al. 2018; <https://doi.org/10.1098/rstb.2019.0094>), however as seen in your Supplementary they are far apart in intron content. Fonticula is fast-evolving (shown in the long branches it has on phylogenies), I wonder if there is any relation with this and the high intron number seen in Parvularia or if it is independent.

- Line 105. The term ‘Genetic turnover’ needs explanation

- Line 152: What do you mean by non-fungal opisthokonts

- Line 156 ‘machine-learning;’ is better for metazoan... I think a bit more clarity and discussion of caveat is needed here. Is some of this metazoan-clarity coming from the fact that we do not have cell/development understanding of protists and fungi in the same way as we do for animals? Will this not mean many aspects of this paper are essentially bias towards finding strong trends in animals away from protists/fungi leaving those two looking like each other. I think this bias has to be true and has to affect the results of the paper. It is all fine, and of course that trend must exist. But the absence in equivalent trends separating the other two (protists and fungi) to the same extent must also partially be true due to absence of understanding in a gene annotation pipelines for these groups. For me this is very clear in Fig 2A. This caveat needs discussing.

- I may have missed it, but it is important to upload a table of origin (DOI, database used, etc.) of the used sequences.

- Line 192 etc . I did not like constantly having to look up what node M4 was etc. Can you not say what they are a bit more (with the bracket (M4) etc).

- line 202, Not sure what you mean by ‘quantitatively more important’. Do you mean the number was higher. Can you unwrap and accurately reword this line with the numbers?

- Lines 476-479 Gene fissions. If only somebody had looked at this in fungi.

<https://doi.org/10.1073/pnas.1210909110>

Referee #4 (Remarks to the Author):

Origin of animal multicellularity is one of the most exciting topics in evolutionary biology. In this manuscript, the authors report sequencing of four new genomes of protists with potentially highly informative positions - branching between animals and their (relatively) close relatives, the fungi.

The detailed comparative analysis of a plethora of opisthokont genomes, including the newly available ones, allows the authors to gain an unprecedented insight into the trajectory of gene gain and loss in both lineages.

The main conclusions are not particularly surprising, given morphologies and lifestyles of extant animals and fungi: while animal genomes increased diversity of genes involved in signaling (in line with their ability to form diverse and complex bodies) the the fungal genomes increased diversity of metabolism-related genes (in line with their ability to use broad diversity of nutrients and produce fascinating chemical compounds). But the manuscript is the first (to my knowledge) to formally demonstrate, and show the evolutionary history of these features.

The most unexpected finding is that these changes were not sudden expansions of gene complexities in these categories at the root of fungi or animals - instead, they were gradual, with the newly sequenced genomes allowing clear insight into the fact that these changes predate emergence of "advanced" animals and fungi. Another intriguing (although perhaps intuitive) point is that fungal genomes are less distinct from protist genomes than animal genomes are (clearly, we animals are very special).

The manuscript is very well written, and I have no doubt will be of great interest to Nature readership. However, especially for a broader audience, it would be great to have a bit more explicit statements and descriptions covering the following points (which might not be immediately clear to non-experts):

Was the last common ancestor of fungi and animals a single- or multicellular organism? What does the literature suggest [Medina et al 2003 wrote "Our data therefore add support to the inference that multicellularity evolved twice in the Opisthokonta, once in the animals and again in the Fungi." - is that still generally considered true?]? What does the analysis presented in the manuscript tell us on that topic? For example, does the reported (and very important) loss of multicellularity-related genes in the early history of the fungal lineage imply that the ur-opisthokont was colonial/multicellular, but this feature was lost along the lineage leading to fungi, before subsequent re-invention of multicellularity?

It would also be helpful to have very brief descriptions of the four newly sequenced protists within the main part of the manuscript - are they always unicellular? Colonial? Ever multicellular?

In general, I felt that the authors were very rigorous in their analyses, but at few sections I was surprised that they considered apparently very minor differences between percentages to be meaningful. For example "In the path to Metazoa, the changes that occurred in the three pre-metazoan ancestors (M1-M3) together account for a larger contribution to shifting the composition of the lineage towards Metazoa-related functional categories than those changes occurred in the metazoan root (3.7% vs 3.5%, Fig. 1A)." Is 3.7% really so different that 3.5%? Am I missing some statistical analysis here?

Similarly, "An inspection of the ancestral contribution to *H. sapiens*' gene content (Extended Data Fig. 5E) illustrates the same trend: gene originations from M4 contributed more to the ancestral repertoire of *H. sapiens*' genes involved in K, T and W (mean 13.9%) than gene originations from M1 to M3 (mean 12.5%). From this, we conclude that gene originations at M4 have been quantitatively more important to functions related to animal multicellularity than gene originations from the preceding holozoan ancestors." Again, 13.9% does not appear to be so different than 12.5%, but I would be happy to be corrected.

Extended data figures would benefit from more extensive legends - it is sometimes hard to understand what some of the numbers (which appear to be randomly appearing among the nodes) mean, for example in Extended Data Fig. 6. and 7.

Very minor - what does the note "Cat y todo junto" in the supplementary table 3 (when referring to Sycon ciliatum data source) mean?

Author Rebuttals to Initial Comments:

Point-by-point answer to reviewers

Referee #1 (Remarks to the Author):

I got this paper and really wanted to like it a lot. I feel that there is almost no methodological problem with the paper - though as you can see, I have a question about what might be circular reasoning in one part. My overall impression is of an interesting paper, but I don't feel particularly surprised by the results. Fungi are metabolic geniuses - this is not surprising, while animals have remodelled their genes extensively, through rounds of duplications, lots of recombination etc. The paper is definitely interesting, but perhaps with a bit of clarifying the small number of points I have to make, it would be easier for me to see how much of a surprising set of data we have here. Personally, I would like to see it published - the thinking is definitely quite novel and the contribution of new key genomes has made this "transition" of animals/fungi very interesting indeed.

We would like to thank referee #1 for the valuable feedback provided on the manuscript. We could not agree more with the fact that the origin of animals and fungi is a very interesting evolutionary question, particularly if it is addressed from a comparative perspective. In this regard, we believe that our manuscript pays a significant contribution to this question, not only by the fact of having characterised the two evolutionary modes that differentiate animals and fungi, but also because we tracked and quantified the gradual emergence of these differences in the phylogeny. We hope that these new genomes will motivate further research on this topic, and hopefully future studies could keep contributing in mitigating the lack of data for those unicellular lineages that branch close to Metazoa and to Fungi in the eukaryotic tree.

I am somewhat at a loss for figure 1 (and its associated discussion) why there are approximately 28 internal branches that are not labeled at all. The authors seem to have picked the branches they wished to label. The ladderizing of the tree is just an arbitrary way of viewing it. Why did the authors focus on approximately 50% of the internal branches and ignore the rest?

The main reason for which we chose the ancestral paths leading to *H. sapiens* and to *N. crassa* is because these species were the ones receiving the highest score by the machine-learning predictors that were trained to detect genomic compositions of functional categories that are characteristic of Metazoa and Fungi, respectively (see 'Hsap' and 'Ncra' in Supplementary Table 5). Furthermore, by choosing the ancestral paths leading to these species (M5-M10, F4-F11), we are covering the largest possible number of phyla within each group and also the largest number of ancestral time-points that could be represented by any ancestral path. Having said that, it is important to note that our analysis (and the interpretation) focus on the earliest events that occurred after the divergence of Opisthokonta to the stem node of both groups (i.e., M1-M4 and F1-F3, and this is why we sequenced novel protists whose phylogenetic position helps to inform our reconstruction of these early steps). Because the genetic changes in M5-M10 and F4-F11 are barely mentioned and have a marginal importance in our manuscript, in agreement with referee #3's suggestions, we moved all the mini-plots corresponding to the ancestral paths to *H. sapiens* and *N. crassa* to Extended Data Fig. 4, which now also includes the mini-plots for

the rest of ancestral nodes in the phylogeny. We believe that these changes allowed us to produce a significantly improved version of Figure 1.

Also for figure 1, could the authors say why they do not represent branch lengths on their tree, only topology?

The same can be said for figure 2 - why are branch lengths not represented on the tree? If it is just aesthetics, then this is probably not a good enough reason.

Both figures present trends in gene content evolution along the Opisthokonta phylogeny. We chose not to represent branch lengths on the tree for two reasons. First, to best use the computational resources at our disposal, we inferred two separate phylogenies (one for Holozoa, another for Holomycota, see Supplementary Information 3-Fig. 1A-B). This approach allowed us to employ complex substitution models, reduce noise from the alignments, and perform multiple analyses to test the robustness of the topologies recovered for Holozoa and for Holomycota. We thus do not have a phylogeny with branch lengths for the whole Opisthokonta. Second, the branch lengths of the trees we reconstructed correspond to the expected number of substitutions per site for the set of markers included in our concatenates and not the geological age of divergence events between taxa. Illustrating gene content evolution lengths along a tree with branch lengths corresponding to the expected number of substitutions per site in our concatenate, is difficult to interpret and potentially misleading. For this reason, we decided to display the topological relationships found as a cladogram, which is the most frequent approach to plot the results of analyses of gene content evolution (e.g., Fernández and Gabaldón 2020, Paps and Holland 2018, Richter et al. 2018, ...). In the caption of the revised Figure 1, we take care to refer the reader to where the Holozoa and Holomycota phylogenies with the inferred branch lengths can be found:

Lines 120-123: “... *The cladogram shown was reconstructed based on the most supported topologies found for Holozoa and for Holomycota in the phylogenetic analyses (Supplementary Information 3).*”

From lines 133 to 149, I am struggling to see how this is not circular reasoning. It seems to me that both the ML classifier and the correspondance analysis- based approach are both identifying fungal- and animal- enriched patterns and then asking whether these patterns are seen in the data. While the fungal-likeness is seen in the unicellular ancestors of animals, there was always going to be a particular branch when it flipped over - it had to flip over at some point.

Thanks for this comment. We see the point and realize we may not have been as clear as we could about the logic of this analysis. The analysis is not circular: first, we used the ML classifier and the correspondence analysis to identify differences in the relative representation of each distinct gene functional category between the extant representatives of Metazoa and Fungi. We then used an independent method, ALE (the gene-tree species-tree reconciliation software), to infer the gene content (and from this, the functional category profiles) of each ancestral node in the Opisthokonta phylogeny. Applying the classifier trained on the extant taxa to these reconstructed ancestral profiles allowed us to determine where, when, and by what process (gradual or punctate) these characteristic

signatures of the modern taxa evolved. The aim of the analysis was not to determine “whether” the patterns are present in the data -we observe clear gene content differences between the animal and fungal tips of the tree- but rather to determine how and when those observed differences emerged. In fact, one of the main results from this analysis is that these patterns did not appear all at once - a “flip” on a single branch - but emerged gradually during the early diversification of Holomycota and (particularly) Holozoa. We have introduced some modifications in our description of the analyses in the main text to fully unpack this logic:

Lines 169-188: *“From an evolutionary perspective, the large genetic differences shown between Metazoa and Fungi might be explained because either both or just one of the two groups experienced substantial genetic changes from their last shared common ancestor. Furthermore, this divergence could either be due to an abrupt genetic turnover in which changes would have occurred specifically in the root of both groups, or by a gradual process in which the preceding ancestors of each group were already accumulating changes in the direction of the differences observed in extant Metazoa and Fungi (Fig. 2A). To distinguish between these alternative scenarios, we took two complementary approaches to reconstruct the tempo and modes of the genetic divergence that occurred. In the first approach, we split the functional categories into two groups based on the results from the multivariate analysis on extant species from Metazoa and Fungi (Fig. 2A): Metazoa-related or Fungi-related. Then, we computed the relative representation of each group of functional categories in every ancestral node of Opisthokonta (Fig. 1A) based on the gene contents inferred with our ancestral reconstruction pipeline (see Methods). In the second approach, we trained a series of machine-learning classifiers to find their own functional category-based definition based on the gene contents from extant Metazoa and Fungi (see Methods). Then, we scored the ancestral nodes -which were not involved in the training of the classifiers- according to how metazoan-like and fungal-like are the relative compositions of functional categories of their inferred gene contents (Extended Data Fig. 4C).”*

Again, if the tree in figure 1A had branch lengths on it, it would be easier to evaluate the significance of this flip-over.

See a couple of points above.

The infographic of figure 4 didn't provide a great deal of enlightenment and I found it difficult to read.

The aim of Figure 4 is to provide a synthesis of the multiple analyses that can be found around the main text and in the supplementary. We have made some changes in the displayed text and in the figure caption; hopefully its interpretation becomes easier now.

The paper reports some key genomes, the statistics are appropriately used and implemented (notwithstanding my question about the circularity of reasoning), the conclusions are robust and the authors do not "oversell" the results and the paper is clearly written.

Referee #2 (Remarks to the Author):

Ocana-Pallares et al.

Divergent genomic trajectories predate the origin of animals and fungi

The authors present a remarkably multifaceted genomics-informed investigation of the evolution of Metazoa and Fungi. These two lineages comprise the best known members of the two main groups of Opisthokonts (animals, fungi and various protists). Focusing on genomes from novel protist lineages that branch 'between' animals and fungi, they provide a window on the patterns and processes that contributed to the emergence these two groups. Some of their conclusions are somewhat confirmatory relative to previous ideas (e.g., role of gene duplication in early animal evolution, more lateral gene transfer in fungi and their protist progenitors relative to on the animal side). But I have never seen this question addressed so comprehensively and definitively, in such a way that makes it so clear what transpired (genetically speaking) on the deepest branches leading to extant Metazoa and Fungi -- and how the seeds were set in so very different ways on the two sides of the Opisthokonta. The really novel parts are where the authors dive into the changes in 'gene functional categories' and '% metabolic genes'. I should think this paper will be very well received and well read.

I also must tip my hat to the authors for presenting an understandably abbreviated version of their truly vast set of results in a way that can be followed. Unlike most papers done at this scale, I found this one very accessible and relatively straightforward to go back and forth between the main text and the supp info (figs and accompanying text). It's not perfect (see below), but a lot of work clearly went into the structure and content of the main text figures.

We thank the referee for their positive insights about our manuscript and also for the feedback provided.

Specific Comments

-The nature of the new genomic data isn't clear enough. It's obviously novel and very important, but what kind of data is it, exactly? Assembled genomes (and presumably transcriptomes)? The abstract says 'four novel genomes', but nowhere else is this clearly addressed and diving into the details in the large amount of supp data / info, it's clear that it's more metagenomic in some cases since these new protists are very difficult to work with.

Thanks for offering us the opportunity to clarify this. As the referee says, the data that we produced for these four species are metagenomes. We have modified the main text to make this information more accessible to the reader without the need of going to the supplementary, as well as to link to the section of the supplementary information where we detail the process by which the data were cleaned, assembled and annotated:

Lines 76-81: *"To improve the limited genome sampling for the protist opisthokont groups³, we sequenced, assembled and annotated the genomes of three filastereans (Ministeria vibrans, Pigoraptor vietnamica, Pigoraptor chileana) and one nucleariid*

(Parvularia atlantis) from metagenomic data produced from cultures of these species (Supplementary Information 1)."

It is understandable that the authors can't easily summarize the nature of the data they acquired and analyzed. But nowhere (that I could find) is there a single table that summarizes what was obtained for each of the four novel protists (it seems to be spread across at least 4 SI Tables). People in the field will want this. They will want to be able to access this information and integrate the new data into their analyses, and assess, e.g., the extent of partiality of the new data (and what they will and will not be able to do with it). I don't question the rigor of their data inclusion / exclusion processes (ESOM maps, etc.). But the authors would do well to provide a single SI table that puts the most important bits all in one place.

Thanks for this suggestion. The manuscript did contain this information (Supplementary Information 1-Fig. 23), but we agree it was difficult to find it since it was not properly referenced in the main text. We have modified the main text to indicate where the reader can find this information.

Lines 95-99: *"The four sequenced genomes present high completeness and contiguity metrics which are in the range of those from the previously sequenced protist opisthokont species (Supplementary Information 1-Fig. 23). With regard to genome size and gene content metrics, the sequenced species are not different from most unicellular eukaryotes and fungi ..."*

-Abstract: "...different preferences for sources of gene gains". Change to 'gene gain/ evolution processes' or something similar? Makes the reader (me at least) think of LGT, which only appears to have been a significant factor on the fungal side.

We have changed "sources of gene gains" for "gene gain mechanisms", which is probably a more accurate term.

-Abstract: "We show that animals took advantage of a tendency of gaining genes of multicellularity-related categories which started in the pre-metazoan ancestors."

Using the term 'multicellularity-related category' made me pause because it sort of sounds as though it's a category that existed prior to multicellularity. I don't have a clean suggestion for a re-phrase, but something along the lines of:

"We show that pre-metazoan ancestors took advantage of a tendency of gaining genes that today are associated with multicellularity".

Thanks for this suggestion. We have modified the text of the abstract so that the previous term is not used.

Lines 41-45: *" ... We show that animals arose only after the accumulation of genes functionally important for their multicellularity, a tendency that began in the pre-metazoan ancestors and later accelerated in the metazoan root ..."*

Additional comments

Line 73: Filasterea (Holozoa) is the closest KNOWN relative group to animals...

Also, 'relative group' doesn't sound quite right.

We have changed it for 'closest known groups to'.

Lines 71-72: “The closest known groups to Metazoa within Holozoa are Choanoflagellata, Filasterea and Teretosporea (Fig. 1D). Within Holomycota, the closest known groups to Fungi ...”

Line 140: this should be tempo and modes, because there doesn't have to be (and there doesn't appear to be) just one mode.

Thanks. We have implemented this change (line 178).

Line 306: “Interestingly, these same categories underwent losses in the pre-fungal ancestors (Fig. 1B), situating the immediate ancestors of Fungi and Metazoa in completely different latent potentials from a genomic perspective.”

I think 'completely different' is too strong here. Rather different? Substantially different?

We agree that “completely” was perhaps too strong, “substantially different” was a good suggestion (line 390), thanks.

Referee #3 (Remarks to the Author):

Review Ocaña-Pallarès et al. 2022

This paper represents a very large-scale analysis investigating the genome evolution between animals and fungi. It is well written and easy to follow (although improvements could be made), it is an interesting read. The methods used by the authors seem strong and repeatable. The supplementary are exhaustive and really detailed. The release of these four new genomes is an important resource for the community. The paper itself does not include any new surprises. It just brings together a lots of thoughts that have been simmering in the field in different sub-disciplines and perspectives. It therefore does justice to a large scale evaluation of these ideas. I think the manuscript is worth publishing and will be a very important reference for the field.

We thank the referee for all the suggestions made which certainly helped us to improve the previous version of the manuscript.

Comment 1:

Is this study focusing only on genomic data? Are the 1,463,920 protein sequences from 83 eukaryotic species all coming from genomes? Did the authors avoid using proteomes coming from transcriptomic data? If so, it should be stated, clarified, and justified why.

We thank the referee for giving us the opportunity to clarify this important point. 76 of the 83 proteomes that were included in the initial dataset come from genomic data. The seven species the proteomes of which were produced from transcriptomic data (two choanoflagellates, five teretosporeans, see Supplementary Table 3) were only incorporated

into our dataset for the first part of our analyses, in particular, to give some extra signal in the phylogenetic reconstruction of the species tree of Opisthokonta (which is one of the inputs for the reconciliation analyses, see euk_db_draft dataset in Supplementary Table 3). They were not included, however, in the gene trees and hence these species were excluded from the dataset used in the gene tree/species tree reconciliation analyses (euk_db_dataset, see Supplementary Table 3). We argue that the inclusion of transcriptomic data could have been particularly problematic for the reconciliation analyses due to the following reasons:

(i) Gene content predictions from transcriptomic tend to present inflated gene counts. For example, the proteomes that were previously produced based solely on transcriptomic data for *P. atlantis* and for *P. vietnamica* and *P. chilleana* include much more sequences (29,620, 46,018 and 37,783) than the proteomes that we predicted from the genome sequences of these species (9,028, 14,822 and 14,510), with the genome-based proteomes showing even better completeness metrics (Supplementary File 1-Fig. 23). Another example: for the 21 proteomes labelled as “Choanoflagellata” in EukProt (<https://github.com/beaplab/EukProt>), the two of them that were produced from genomic data (*Monosiga brevicollis* and *Salpingoeca rosetta*) present 9,203 and 11,731 sequences, whereas the FASTA files for the other choanoflagellates, which were produced solely based on transcriptomic data, range from 61,053 to 18,816 sequences and have 30,395 sequences on average. Inflated gene counts are expected to produce an excess of erroneous ancestral duplication inferences in the reconciliations. (ii) The absence of those unexpressed genes in the transcriptomic data may be confused by gene losses in the reconciliation. (iii) Transcriptomes are harder to decontaminate due to the lack of genomic context information regarding neighbouring genes, intron sequences, or compositional features of the coding sequence (this information was particularly valuable for us to decontaminate the genomes of the four species sequenced). (iv) Partial isoforms in transcriptomes can lead to the prediction of incomplete sequences, and these can lead to inferences of false gene fusion events by the software that we used for this purpose (CompositeSearch, Pathmanathan et al. 2018). (v) A strength of the method used for the reconciliation analyses (ALE) is that some parameters, such as the duplication and loss rates, can be inferred from the data. The usage of accurate gene contents was thus particularly important for us.

We have incorporated a shorter version of this piece of text into the Methods section (lines 639-653).

I personally do not see a reason (even when transcriptome data can be limited) for not using proteomes coming transcriptomes (if required) and I cannot see a clearer example than using the only available transcriptome from the key group the aphelids (e.g. Paraphelidium tribonemae) for these analyses. Using this transcriptome would certainly provide a control for the trends they report at multiple nodes.

To this point it seems clear that aphelids are the sister lineage to Fungi (making Opisthosporidia paraphyletic) (Torruella et al. 2018; <https://doi.org/10.1038/s42003-018-0235-z> and Tikhonenkov et al. 2020; <https://doi.org/10.1016/j.cub.2020.08.061>). Thus, given their phylogenetic position is surprising that the authors make no mention of aphelids in the text.

What is more, beyond their newly proposed phylogenetic position it has been shown that some hyphal multicellularity related categories (e.g. cell wall biogenesis/remodeling' and 'transcriptional regulation') evolved in the ancestor of aphelids and fungi (Galindo et al. 2021; <https://doi.org/10.1038/s41467-021-25308-w>). I understand that remaking all analysis for only one transcriptome can be troublesome, however, aphelids are in a key position to understand the picture the authors are trying to draw.

For example: Given your results the fungal root (F3) is the node with the biggest with the largest fraction of genes gains and metabolic expansion. However, part of this fraction could be explained by the non-present aphelid node.

The same could be said about *Olpidium bornovanus*, in this case there is both a genome and transcriptome which have been recently obtained (Chang et al. 2021; <https://doi.org/10.1038/s41598-021-82607-4>) showing that *Olpidium* falls within its on phylum which branches right at the phylogenetic border between zoosporic and non-zoosporic fungi. Adding this fungus to the analysis could have sharpen these results.

P. tribonemae (aphelid) and *O. bornovanus* were not included into our original dataset because their gene contents were not available at the time we designed the taxon sampling for this project. Moreover, the aphelid data comes from a transcriptome (see above for a justification on why gene content predictions coming from transcriptomes could be problematic for our study). Notwithstanding this, based on the referee's suggestion, we decided to do an additional supplementary analysis with a dataset that incorporates *P. tribonemae* (aphelid) and also *O. bornovanus*. We decided to put the results from this additional supplementary analysis in the Supplementary (Supplementary Figure 1) and maintain the results from the original dataset in the main text, which are consistent in the sense that they are only based on data coming from genomes.

The results from the additional supplementary analysis confirm our previous findings and also increase the support for some of our inferences. On the one hand, the new results confirm our previous findings because all trends of change at the level of genomic composition of functional categories (Suppl Fig. 1A-B), net gains and losses of functional categories (Suppl Fig. 1C), and net gains and losses of metabolic genes (Suppl Fig. 1D) are consistent at node-level between the original and the new dataset; only the ancestral nodes that are located after the positions in which the newly added taxa branch experienced some changes (this pattern is particularly evident in panels C and D). On the other hand, the new results suggest that the genetic changes that occurred at the preceding ancestors of Fungi would have led to a greater compositional shift towards Fungi-related functional categories than what we observed in the results from the original dataset (0.9%+0.9% for F1 and F2 in the original dataset; 1.1%+1.1%+0.6% for F1, F2 and F2.5 in the new dataset, with F2.5 corresponding to the last common ancestor of Fungi and Aphelida; Supplementary Fig. 1A-B). Interestingly, the new dataset suggests that approximately half of the net gains that we detected at the root of Fungi (F3) in the original dataset would have occurred before the origin of Fungi, in particular before the divergence of Fungi and Aphelida (i.e., at F2.5). Notwithstanding this, the root of Fungi still appears in the new dataset as a node that would have experienced net gene gains in the vast majority of functional categories including the metabolic ones. We have included a reference to this result in the main text:

Lines 282-290: "... the metazoan root experienced a net loss of metabolic genes despite this node presenting an overall net gain of gene content (Fig. 1B), while the fungal root experienced net metabolic gene gains (Extended Data Fig. 8A). (Note that an additional supplementary analysis with a dataset that includes transcriptomic data from the aphelid *Paraphelidium tribonemae*⁹, which is the closest known group to Fungi, suggests that half of the net gene gains originally detected at the fungal root, including the metabolic ones, could have also predated the origin of Fungi, see Supplementary Figure 1)."

Comment 2:

Given its importance for multicellularity (at least in multicellular fruiting-body producing fungi): Is there any reason why cell adhesion, or adhesion-related genes are not explored (or at least not shown) in the functional categories? If adhesion is by any chance included within other functional category (e.g. cytoskeleton) It should be address on its own, since as seen in previous studies (Kiss et al. 2019; <https://doi.org/10.1038/s41467-019-12085-w> and Galindo et al. 2021; <https://doi.org/10.1038/s41467-021-25308-w> (Supplementary Fig 13)) adhesion-related genes expanded in complex multicellular fungi (mainly Dikarya), thus having a crucial role in the current (fruiting body-producing) hyphal-based multicellularity.

We explored the evolution of multicellularity in animals because it is strictly related to the origin of this group. Although the evolution of complex multicellularity (CM) in Fungi is also a fascinating topic, CM is not a core feature of Fungi. It only evolved secondarily in some specific groups which are phylogenetically distant from the organisms for which we produced new genomic data. Indeed, our dataset was particularly designed to explore those early genetic changes that occurred since the divergence of the last common ancestor of animals and fungi to the origin of both groups. Notwithstanding this, we agree with the referee that this is an interesting topic, and that our analyses provide a framework for investigating it. For this reason, we have performed additional analyses, reported in a new supplementary section (Supplementary Information 4), in which we explore the origin/s of CM in Fungi in an "unsupervised" manner, i.e., by using statistical tools that find global trends of change by evaluating the gene content on its whole, which is consistent with the methodological approach that we have followed along the manuscript.

We believe that the results from this new supplementary complements some previous studies in which this question has been explored by reconstructing the evolution of a subset of *a priori* selected gene families. In brief, we let the RandomForest classifier find a subset of gene families, the copy number distribution of which correlates with extant CM fungi taxa. Then, we checked the copy number distribution of these families in the ancestral nodes. While the largest increments in copy number for these families coincide with the root of Pezizomycotina and Agaricomycotina (the two nodes that evolved the most complex CM in Fungi), we also detected copy number increments at the root of Dikarya (the last common ancestor of Pezizomycotina and Agaricomycotina) and in the ancestral node of Dikarya+Mucoromycotina (which corresponds to the last common ancestor of all fungal groups for which fruiting-body/fruiting-body-like morphologies have been described). Our results thus agree with what is probably the most prevalent hypothesis for CM origin/s in Fungi, which is that the few fungal groups that present CM evolved it convergently, perhaps not from scratch but from some rudimentary version of CM that could have been present in the last common ancestor of multicellular fungi.

Comment 3:

As mentioned in line 151 it is not surprising that the overall composition of the Fungi-category established here is more protist-like than the Metazoan-category, however, why this is happening is not explored enough.

This result fits within the fact that many of the Fungal phyla (in particular the early diverged lineages) are in fact unicellular organisms (Chytriomycota, Blastocladiomycota, Zoopagomycota). This is one of the main differences between Metazoan and Fungi, in Metazoa multicellularity was present in their root, and in fungi evolved later. The presence of unicellular organisms within fungi needs to be highlighted even more since it justifies the observed patterns.

The definition of fungi is to this day problematic, and many fungal clades could be argued to be protist clades (there is a continuity of zoosporic unicellular organisms from apheleids and through chytrids, sanchytrids-blastoclads and Olpidium) (Richards et al. 2017; 10.1128/microbiolspec.FUNK-0044-2017). I think the point 'what is a fungus is important and should be considered from line 226-228

We thank the referee for pointing these things out. We added the following sentence in the final section of the manuscript in which we discuss our results.

Lines 363-371: "Given that the latter result is independent of gene function annotation, Metazoa being more differentiated than Fungi from the rest of opisthokonts from a gene content perspective is robust to potential inequalities that may exist between groups at the level of biological knowledge or in the availability of functional information. This indeed agrees with the fact that there are more evident morphological discontinuities between protists and animals than between protists and some groups of Fungi⁸. Neither the hypha nor the cell wall characteristic of Fungi, which is also present in some of their protist relatives, are fungal synapomorphies⁸. Only the abandonment of phagotrophy for an osmotrophic lifestyle seems to be a common although not exclusive feature of Fungi³²."

Furthermore, we have also included a sentence at the beginning of the main text to clarify our taxonomic definition of Fungi.

Lines 72-76: "Within Holomycota, the closest known groups to Fungi (here defined as the least inclusive clade including Chytridiomycota and Blastocladiomycota based on the absence of phagotrophy in all the members of this clade^{8,9}) are Opisthosporidia (a paraphyletic group^{8,10} which in our genomic dataset is represented by *Rozella allomycis* and *Mitosporidium daphniae* -RM clade-) and Nucleariidea (Fig. 1D)."

Comment 4:

It is interesting how the authors show that the biggest shifts towards Fungi-related functional categories occurred in the nodes F5 and F7, however it has not been discussed why. Loss of flagellum? Gain of aerial spores? Gain of hyphal traits?

F5 is the node in which fungi lost their flagellated life-stages (zoospores) and started to have septate hypha-like structures, and F7 is the shift towards Dikarya, in which “true” multicellular fruit-body producing fungi evolved. I think these large evolutionary innovations fit perfectly into their results. Please highlight it.

We did not discuss the findings in the nodes that follow the fungal stem node given that we are mainly interested in the genetic changes that precede and coincide with the origin of fungi (same for Metazoa). However, we agree that the information pointed by the referee could be of interest to the reader, and hence we included it in the main text.

Lines 262-269: *“The two fungal nodes that present the largest compositional shift towards Fungi-related functional categories (see Fig. 1D) are, on the one hand, the stem node of Dikarya (Ascomycota+Basidiomycota) (+1.9%), which experienced genetic changes that could have predisposed the evolution of complex multicellularity in some members of this group (see Supplementary Information 4), and on the other, the last common ancestor of Zoopagomycota, Mucoromycotina and Dikarya (+1.5%), which experienced important morphological adaptations such as the ancestral loss of the flagellum that is characteristic of most fungal groups²³”*

Comment 5

I am struggling with figure 1. I find the mini graphs impenetrable. Do you need MG6-M10. Given the war zone that is the early animal phylogeny do you need M5. Can these be in supplementary materials? Likewise do you need F5-F11. Then you could have less mini graphs, make them bigger and order them around the key nodes with the tree on its ‘back’. I think that would be more approachable.

Thanks for this suggestion. We believe it helped us to significantly improve Figure 1. The current version includes the mini-plots for the pre-metazoan and pre-fungal ancestor (M1-M3 and F1-F2) and for the stem nodes of both groups (M4 and F3), which now are shown with higher resolution. The other mini-plots have been moved to Extended Data Fig. 4 as their relevance for the main text is residual. We have also incorporated the mini-plots for the rest of the ancestral nodes into this supplementary figure.

Minor comments

- *Line 48 – define Fusions? I think you mean gene fusions*

Yes, we have replaced it by gene fusions.

- *Line 51 – why reduced HGT in metazoans. Weissman barrier? Is this all metazoans? Given the controversy of the animal HGT stories, please be specific. I think you can be and still fit it in an abstract.*

It has been hypothesised for a long time that HGT should be a less important source of gains in Metazoa due that germ-line isolation (Weissman barrier) is a general feature of this eukaryotic group. Our results from the gene tree-species tree reconciliation analyses are

concordant with the prediction of this hypothesis. We extended the abstract to incorporate a mention of it.

- *Line 74: Calling Fungi + “Opisthosporidia” the ‘Fungi sensu lato’ may be a big stretch, since Microsporidia and rozellids are far from the canonical definition of fungi. Please refer to Nuclearia+”Opisthosporidia”+Aphelids+Fungi as Holomycota, and only refer to true fungi (Chytridiomycota and on) as fungi. Also refer to “Opisthosporidia” using quotes, given its paraphyletic nature.*

We agree with the definition of Fungi proposed by the referee. We, however, prefer not to use the term Opisthosporidia given its paraphyletic nature. For this reason, we have changed the term Opisthosporidia (we only mention it at the beginning of the main text, see below) for the term R+M clade, which is an abbreviation for the clade defined by *Rozella allomycis* and *Mitosporidium daphniae*, the two opisthosporidians included in our original dataset (i.e., the pre-aphelid one).

- *Overall, I am missing in the introduction a paragraph in which the authors define (as for them and as for this paper) what is fungi and what is Holomycota, since terms like (non-fungal opisthokonts; line 152-153) are terms that can create confusion.*

We now define Holomycota and Fungi at the beginning of the main text:

*Lines 72-76: “Within Holomycota, the closest known groups to Fungi (here defined as the least inclusive clade including Chytridiomycota and Blastocladiomycota based on the absence of phagotrophy in all the members of this clade^{8,9}) are Opisthosporidia (a paraphyletic group^{8,10} which in our genomic dataset is represented by *Rozella allomycis* and *Mitosporidium daphniae* -RM clade-) and Nucleariidea (Fig. 1D).”*

- *Line 84-89: It is incredibly interesting the super-high intron rate of Parvularia. I would highlight even more that Parvularia has by far the highest intron numbers of any eukaryote but at the same time its introns are one of the shortest. What are these introns? Any further exploration? Are they palindromic repetitive sequences? Short tandem repeats? Do they come from transposons? I think that it should definitely be explored (maybe in a follow up paper).*

We were also surprised by this finding. Regarding the nature of these introns, we ran RepeatModeler and RepeatMasker to check whether introns coincide with sequences identified as low-complexity or transposable elements, but this does not seem to be the case. Also, we did not find the intron sequences sharing significant similarities between them. Because, after all, this genome was produced to increase the availability of genomic data in early-Opisthokonta, we preferred to leave this result as a curiosity, and we will be happy to see future studies exploring this question in detail.

- *We know Parvularia and Fonticula share the same clade within nucleariids (Galindo et al. 2018; <https://doi.org/10.1098/rstb.2019.0094>), however as seen in your Supplementary they are far apart in intron content. Fonticula is fast-evolving (shown in the long branches it has on phylogenies), I wonder if there is any relation with this and the high intron number seen in Parvularia or if it is independent.*

Parvularia is a short branch compared to *Fonticula*, and it is true that there are long branches from distinct eukaryotic groups which tend to be poor in introns (e.g., *Saccharomyces*, *Corallochytrium limacisporum*, *Entamoeba histolytica*, ...). However, other species are also long-branches and present high intron contents, such as the choanoflagellate *Salpingoeca rosetta* (~7.44 introns per gene). On the other hand, *Fonticula alba* is a long branch but still presents a moderate number of introns (3.73 per gene). It would be interesting to explore whether there is a correlation between fast-evolving rates and the intronization of a genome. Notwithstanding whether this correlation exists, the fact that *Parvularia* presents a small genome size and its introns are very short suggests that this lineage evolves under a selective pressure favouring genome streamlining. If so, we could speculate that the presence of introns in its genome, providing that they are unrelated to any transposable element, could be explained because introns are playing some important role in this species (perhaps at the level of controlling alternative splicing?). Overall, it would be great to see future studies exploring this anomalous case. Unfortunately, the fact that *Parvularia* is a poorly characterised species and that in culture, it grows with various contaminant bacterial species is certainly a limitation for potential functional studies in this species that could help clarify this.

- Line 105. The term 'Genetic turnover' needs explanation

Thanks for this suggestion. We have included a definition of what we mean by genetic turnover in the main text:

Lines 136-139: "*a substantial genetic turnover occurred (i.e., the remodeling of the gene content as a result of gene gains and losses, with gains including the origination of novel gene families and the expansion of ancestral families).*"

- Line 152: *What do you mean by non-fungal opisthokonts*

We have replaced "non-fungal opisthokonts" by "non-metazoan and non-fungal opisthokont groups".

Lines 190-193: "*Not surprisingly, Fungi-related functional categories are more represented in Fungi (particularly in Basidiomycota and Ascomycota groups), but for most of the non-metazoan and non-fungal opisthokont groups, the relative genomic representation of functional categories is more Fungi-like than Metazoa-like (Fig. 1A).*"

- Line 156 'machine-learning;' is better for metazoan... I think a bit more clarity and discussion of caveat is needed here. Is some of this metazoan-clarity coming from the fact that we do not have cell/development understanding of protists and fungi in the same way as we do for animals? Will this not mean many aspects of this paper are essentially bias towards finding strong trends in animals away from protists/fungi leaving those two looking like each other. I think this bias has to be true and has to affect the results of the paper. It is all fine, and of course that trend must exist. But the absence in equivalent trends separating the other two (protists and fungi) to the same extent must also partially be true due to absence of understanding in a gene annotation pipelines for these groups. For me this is very clear in Fig 2A. This caveat needs discussing.

We thank the referee for bringing up this topic, and we agree that it is important to discuss it. Differences at the level of biological knowledge and gene functional information between eukaryotes exist, and our work, as well as any other paneukaryotic comparative genomics paper that goes beyond those few clades that are populated by model organisms is limited by this. Notwithstanding this, the result that animals are more different to protists than Fungi, which is one of our main conclusions, is robust to this limitation. We have checked for differences at the level of gene family content between opisthokonts. Note that this is independent of gene function annotation. It is only about the number of copies that every gene family (orthogroups) has in each species. When clustering the opisthokonts according to this information, Metazoa appears in a separate cluster from the rest of the opisthokonts, which include Fungi and the protist groups (Extended Data Fig. 5G). We have included this information in the discussion at the end of the main text.

Lines 360-366: *"metazoan gene contents are more diverged than the fungal ones from those of the other opisthokonts both at the broad-scale functional level and at the gene family content level (Extended Data Fig. 5C and G). Given that the latter result is independent of gene function annotation, Metazoa being more differentiated than Fungi from the rest of opisthokonts from a gene content perspective is robust to potential inequalities that may exist between groups at the level of biological knowledge or in the availability of functional information."*

- *I may have missed it, but it is important to upload a table of origin (DOI, database used, etc.) of the used sequences.*

This information has been made available in Supplementary Table 3.

- *Line 192 etc . I did not like constantly having to look up what node M4 was etc. Can you not say what they are a bit more (with the bracket (M4) etc).*

In the current version of the manuscript, we have minimised the usage of the abbreviation M4 in favour of the terms "metazoan root" or "metazoan stem node". Also, Fig. 1 currently shows in a much more evident way what we mean by M1-M3 and F1-F3.

- line 202, Not sure what you mean by 'quantitively more important'. Do you mean the number was higher. Can you unwrap and accurately reword this line with the numbers?
- *Lines 476-479 Gene fissions. If only somebody had looked at this in fungi.* <https://doi.org/10.1073/pnas.1210909110>

It is interesting that fusions were found in a much lower ratio than fission in fungi (1:1.746). We incorporated this reference into the manuscript, as this finding agrees with the fact that fusions would perhaps be less prevalent in Fungi than expected, at least compared to Metazoa.

Lines 327-328: *"Fusions being less prevalent in Fungi agrees with a previous study that reported a particularly low rate of fusions compared to fissions³²"*

Referee #4 (Remarks to the Author):

Origin of animal multicellularity is one of the most exciting topics in evolutionary biology. In this manuscript, the authors report sequencing of four new genomes of protists with potentially highly informative positions - branching between animals and their (relatively) close relatives, the fungi. The detailed comparative analysis of a plethora of opisthokont genomes, including the newly available ones, allows the authors to gain an unprecedented insight into the trajectory of gene gain and loss in both lineages.

The main conclusions are not particularly surprising, given morphologies and lifestyles of extant animals and fungi: while animal genomes increased diversity of genes involved in signaling (in line with their ability to form diverse and complex bodies) the the fungal genomes increased diversity of metabolism-related genes (in line with their ability to use broad diversity of nutrients and produce fascinating chemical compounds). But the manuscript is the first (to my knowledge) to formally demonstrate, and show the evolutionary history of these features.

The most unexpected finding is that these changes were not sudden expansions of gene complexities in these categories at the root of fungi or animals - instead, they were gradual, with the newly sequenced genomes allowing clear insight into the fact that these changes predate emergence of "advanced" animals and fungi. Another intriguing (although perhaps intuitive) point is that fungal genomes are less distinct from protist genomes than animal genomes are (clearly, we animals are very special).

The manuscript is very well written, and I have no doubt will be of great interest to Nature readership.

We thank the referee for their positive comments about the manuscript and also for helping us in improving the manuscript.

However, especially for a broader audience, it would be great to have a bit more explicit statements and descriptions covering the following points (which might not be immediately clear to non-experts):

Was the last common ancestor of fungi and animals a single- or multicellular organism? What does the literature suggest [Medina et al 2003 wrote "Our data therefore add support to the inference that multicellularity evolved twice in the Opisthokonta, once in the animals and again in the Fungi." - is that still generally considered true?]? What does the analysis presented in the manuscript tell us on that topic? For example, does the reported (and very important) loss of multicellularity-related genes in the early history of the fungal lineage imply that the ur-opisthokont was colonial/multicellular, but this feature was lost along the lineage leading to fungi, before subsequent re-invention of multicellularity?

It would also be helpful to have very brief descriptions of the four newly sequenced protists within the main part of the manuscript - are they always unicellular? Colonial? Ever multicellular?

We agree with the referee that the origin of multicellularity is a particularly fascinating topic which perhaps deserved more attention in our previous version of the manuscript. It is a complex topic, a thorough exploration of which under the light of the new data would probably deserve a separate manuscript. Notwithstanding this, we agree with the referee that there could be a significant fraction of the audience that could be interested in learning more about this topic without the need of looking for some of the notable references that have been published recently (e.g., <https://doi.org/10.1098/rsob.200359>, <https://doi.org/10.1016/j.fbr.2020.07.002>). For this reason, we have produced a whole new

supplementary material about it (Supplementary Information 4) in which we offer to the reader the possibility to learn about the different forms of multicellularity that can be found across Opisthokonta (in Metazoa and in Fungi, but also the simpler forms of multicellularity that can be found in some protist relatives of both groups, including the organisms that we sequenced). We also mention in the supplementary that the last common ancestor of Opisthokonta could have presented a rudimentary version of multicellularity, or at least the potential to evolve it. In fact, we agree that the genetic changes that we describe to have occurred in the path to Metazoa probably predisposed the emergence of a completely-multicellular group such as animals, while the other way around in the case of Fungi, as we mention in the discussion.

Lines 384-390: *"In particular, the genetic changes at the metazoan root represent an acceleration of a tendency which was already ongoing in the pre-metazoan ancestors to accumulate genes of functional categories that are important for animal multicellularity (Extended Data Fig. 5F). Interestingly, these same categories underwent losses in the pre-fungal ancestors (Fig. 1B), situating the immediate ancestors of Fungi and Metazoa in substantially different latent potentials from a genomic perspective"*

On the other hand, although nowadays there is a clear consensus about the fact that (some specific groups of) Fungi took a completely different path than Metazoa to evolve multicellularity (e.g., <https://doi.org/10.1016/j.fbr.2020.07.002>), it is unclear how many times complex multicellularity (CM) evolved in Fungi. We have also explored this particular question in Supplementary Information 4, and our results agree with what is probably the most prevalent hypothesis for CM origin/s in Fungi, which is that the few fungal groups that present CM evolved it convergently, perhaps not from scratch but from some rudimentary version of CM that could have been present in the last common ancestor of multicellular fungi. We have incorporated a reference to this new supplementary in a section of the main text in which we mention the most important differences between Metazoa and Fungi at the level of multicellularity:

Lines 371-375: *"While animals distinguish from protists from the fact that all of them are multicellular, in Fungi, complex multicellularity is probably the outcome of convergent evolution as it is only found in some particular groups which present important differences in the genetic contents involved on it¹⁷ (see Supplementary Information 4 for further information on the evolution of multicellularity in Opisthokonta and in Fungi)."*

In general, I felt that the authors were very rigorous in their analyses, but at few sections I was surprised that they considered apparently very minor differences between percentages to be meaningful. For example "In the path to Metazoa, the changes that occurred in the three pre-metazoan ancestors (M1-M3) together account for a larger contribution to shifting the composition of the lineage towards Metazoa-related functional categories than those changes occurred in the metazoan root (3.7% vs 3.5%, Fig. 1A)." Is 3.7% really so different that 3.5%? Am I missing some statistical analysis here?

Similarly, "An inspection of the ancestral contribution to H. sapiens' gene content (Extended Data Fig. 5E) illustrates the same trend: gene originations from M4 contributed more to the ancestral repertoire of H. sapiens' genes involved in K, T and W (mean 13.9%) than gene originations from M1 to M3 (mean 12.5%). From this, we conclude that gene originations at M4 have been quantitatively more important to functions related to animal multicellularity

than gene originations from the preceding holozoan ancestors." Again, 13.9% does not appear to be so different than 12.5%, but I would be happy to be corrected.

We agree with the referee that the differences between the numbers compared are not very great, so we have decided to rephrase it:

Lines 206-210: *"In the path to Metazoa, the changes that occurred in the three pre-metazoan ancestors (M1-M3) together account for a contribution of a similar magnitude to shifting the composition of the lineage towards Metazoa-related functional categories than those changes occurred in the metazoan root (3.7% vs 3.5%, Fig. 1D)."*

Lines 239-244: *"An inspection of the ancestral contribution to H. sapiens' gene content (Extended Data Fig. 5E) illustrates the same trend: genes from families originated in M4, a single ancestral node, contributed in a similar extent to the ancestral repertoire of H. sapiens' genes involved in K, T and W (mean 13.9%) than genes from families originated in the three pre-metazoan ancestral nodes (M1-M3) (mean 12.5%)."*

Extended data figures would benefit from more extensive legends - it is sometimes hard to understand what some of the numbers (which appear to be randomly appearing among the nodes) mean, for example in Extended Data Fig. 6. and 7.

We have improved the figure captions for these and other extended data figures.

Very minor - what does the note "Cat y todo junto" in the supplementary table 3 (when referring to Sycon ciliatum data source) mean?

We have deleted it as these were old notes which are not meaningful, thanks for pointing it out.

Reviewer Reports on the First Revision:

Referees' comments:

Referee #1 (Remarks to the Author):

I struggled to read the paper because there was no submitted version with the "changes accepted". Therefore, it was more difficult than I expected to do this review.

Nonetheless, I feel this is a significantly strengthened paper. I have read the rebuttal document as it related to my original review and I understand the authors perspective. I also appreciate that they have done a good job in making the logic clearer and the paper as a whole is clearer. I also think that in the process of clarifying some of the points, they have made the argument in favour of publishing the paper stronger.

I would be very happy to see the paper published in its current form.

Referee #2 (Remarks to the Author):

I am satisfied with the thoughtful and thorough revision of the manuscript.

Referee #3 (Remarks to the Author):

I think the authors have done a good job at revising this paper. Although I think they missed one minor point from our Review.

- line 202, Not sure what you mean by 'quantitatively more important'. Do you mean the number was higher. Can you unwrap and accurately reword this line with the numbers?

Referee #4 (Remarks to the Author):

I am happy with the changes made in response to my (and other referees') comments.

Author Rebuttals to First Revision:

Point-by-point answer to reviewers

Referees' comments:

Referee #1 (Remarks to the Author):

I struggled to read the paper because there was no submitted version with the "changes accepted". Therefore, it was more difficult than I expected to do this review.

Nonetheless, I feel this is a significantly strengthened paper. I have read the rebuttal document as it related to my original review and I understand the authors perspective. I also appreciate that they have done a good job in making the logic clearer and the paper as a whole is clearer. I also think that in the process of clarifying some of the points, they have made the argument in favour of publishing the paper stronger.

I would be very happy to see the paper published in its current form.

Referee #2 (Remarks to the Author):

I am satisfied with the thoughtful and thorough revision of the manuscript.

Referee #3 (Remarks to the Author):

I think the authors have done a good job at revising this paper. Although I think they missed one minor point from our Review.

• *line 202, Not sure what you mean by 'quantitatively more important'. Do you mean the number was higher. Can you unwrap and accurately reword this line with the numbers?*

Thanks for pointing this out. We have re-written the sentence (lines 218-220):

Before: "From this, we conclude that gene originations at M4 have been quantitatively more important to functional categories ..."

After: "From this, we conclude that gene originations at M4 have been quantitatively more important (13.9% vs 12.5%) to functional categories ..."

Referee #4 (Remarks to the Author):

I am happy with the changes made in response to my (and other referees') comments.